# How High is 'High'? Rethinking the Roles of Dimensionality in Topological Data Analysis and Manifold Learning

Hannah Sansford [1]  Nick Whiteley [1]  Patrick Rubin-Delanchy [2]

## Abstract

High-dimensionality of data is often regarded as a fundamental statistical impediment in Machine Learning and AI. The purpose of this paper is to clarify, on the contrary, when and how high-dimensionality may be beneficial. In the setting of a general random function model of data we delineate between three notions of dimensionality: *effective dimension* $p_{\text{eff}}$, measuring total variability across feature directions; *correlation rank* $r$, measuring functional complexity across samples; and *latent intrinsic dimension* $d$ of manifold structure hidden in data. Via a generalized Hanson-Wright inequality, we show that increasing $p_{\text{eff}}$ drives a *blessing of dimensionality* phenomenon, whereby data dot-products concentrate about their expectations. In turn, we show that, under mild continuity assumptions (ensuring that features bring additional information as dimension grows), persistence diagrams recover latent homology when $p_{\text{eff}} \in \omega(\log n)$ as $n \to \infty$. Informed by our theory, we revisit the ground-breaking neuroscience discovery of toroidal structure in grid-cell activity made by Gardner et al. (2022): our findings provide the first empirical evidence that this structure is *isometric* to a flat torus model of physical space, suggesting that grid cell activity conveys a geometrically faithful representation of the real world.

## 1. Introduction

**Data geometry in Machine Learning.** Consider a data set of $n$ samples and $p$ features, $\mathbf{Y}_1, \ldots, \mathbf{Y}_n \in \mathbb{R}^p$. A wide range of algorithms, models, supervised and unsupervised learning methods process such data by taking as inputs the pairwise Euclidean distances, dot-products or cosine similarities,

$$\|\mathbf{Y}_i - \mathbf{Y}_j\|^2 = \|\mathbf{Y}_i\|^2 + \|\mathbf{Y}_j\|^2 - 2\mathbf{Y}_i \cdot \mathbf{Y}_j,$$
$$\text{CosSim}(\mathbf{Y}_i, \mathbf{Y}_j) = \frac{\mathbf{Y}_i \cdot \mathbf{Y}_j}{\|\mathbf{Y}_i\|\|\mathbf{Y}_j\|}. \tag{1}$$

The collection of these pairwise quantities across $i, j = 1, \ldots, n$ conveys the geometric shape of the point-cloud $\mathcal{Y}_n := \{\mathbf{Y}_1, \ldots, \mathbf{Y}_n\} \subset \mathbb{R}^p$, and is the input to most dimension reduction and manifold learning methods, ranging from Classical Multidimensional Scaling (Torgerson, 1952), to Kernel PCA (Schölkopf et al., 1997), Isomap (Tenenbaum et al., 2000), $t$-SNE (Van der Maaten & Hinton, 2008) and UMAP (McInnes et al., 2018), which are hugely popular for data visualisation, as well as spectral clustering (Von Luxburg, 2007) (via a kernel function) and hierarchical, agglomerative clustering methods such as UPGMA (Sokal, 1958; Gray et al., 2023). The set of all pairwise distances is also the input to Persistent Homology techniques in topological data analysis (TDA) (Edelsbrunner et al., 2002; Zomorodian & Carlsson, 2004; Chazal et al., 2009; Chazal & Michel, 2021). Pairwise distances or dot-products are the input to classical supervised learning methods such as kernel methods and nearest-neighbour methods (Schölkopf & Smola, 2002), and dot-products between training and test feature vectors define predictions in linear regression.

**The disconnect between reality and existing statistical theory of data geometry.** In this work we present new insights into statistical properties of the quantities in (1), allowing us to explain how they convey latent geometry of the data-generating mechanism. In doing so we address two mismatches between the prominent statistical theory of data geometry and the realities of data analysis. Firstly, the popular mathematical view of a mean-zero, identity-covariance random vector is that, in high dimensions, its distribution is concentrated near the surface of a hypersphere, and that i.i.d. copies of such a vector are close to orthogonal with high probability, see Hall et al. (2005), the textbook of Hastie et al. (2009)[Sec. 2.5], and Cai et al. (2013) for refined analysis of similar i.i.d. setups. This rather degenerate limiting geometry does not seem expressive enough to accurately represent data in practice; it is incompatible with the widely

[1] School of Mathematics, University of Bristol, UK [2] School of Mathematics, University of Edinburgh, UK. Correspondence to: Nick Whiteley <nick.whiteley@bristol.ac.uk>.

accepted *Manifold Hypothesis* (Bengio et al., 2013; Fefferman et al., 2016), which is the premise of manifold learning and nonlinear dimension reduction, asserting that nominally high-dimensional data actually lie on a low-dimensional set embedded in high-dimensional space. Secondly, much existing theory, e.g., Hall et al. (2005); Ahn et al. (2007); Shen et al. (2016); Aoshima et al. (2018) focuses on the regime $p/n \to \infty$ with which the phrase "high-dimension low sample size asymptotics" (HDLSS), and more broadly "high-dimensional data", are typically associated. This existing theory may therefore seem irrelevant in the many practical situations where $p \leq n$.

**Contributions.** Our main contributions are as follows:

- In Section 2 we present a generalised Hanson-Wright (GHW) inequality which quantifies the concentration behaviour of dot-products between possibly dependent random vectors with sub-Gaussian entries, and identify our first notion of dimension: *effective dimension* $p_{\text{eff}}$.
- In Section 3 we identify a second notion of dimension: *correlation rank*, and use our GHW inequality to explain how latent topological structure emerges from a random function model of data (from Whiteley et al. (2026)) as $p_{\text{eff}}$ grows; we show under sub-Gaussian assumptions that $p/n \to \infty$ is not necessary, rather $p_{\text{eff}} \in \omega(\log n)$ as $n \to \infty$ suffices.
- In Section 4 we consider consistency of TDA persistence diagrams in the regime $p_{\text{eff}}/\log n \to \infty$ as $n \to \infty$.
- In Section 5 we identify a third notion of dimension: *latent intrinsic dimension*, and discuss how isometry between observed and latent geometry can be measured in practice.
- In Section 6 we show how our theory manifests in both simulated and real data, including the groundbreaking neuroscience study of Gardner et al. (2022), where latent spatial structure is known to exist.

Section 3 is directly influenced by geometric perspectives of Whiteley et al. (2021; 2026); Gray et al. (2023) and in particular the insightful statistical treatment of the manifold hypothesis in Whiteley et al. (2026), but our main theoretical results (theorems 2.1, 3.1 and Proposition 3.2) substantially extend and refine some of their results; we allow more general dependence structure across features and exploit sub-Gaussianity. An extended discussion of related work is given in the supplementary material, along with all proofs and additional numerical experiments.

**Notation and conventions.** We write $[n] := \{1, \ldots, n\}$ and $\ell_2$ for the set of square-summable real sequences $\{x = (x_1, x_2, \ldots) : \|x\| := (\sum_k |x_k|^2)^{1/2} < \infty\}$. For two non-negative sequences $(a_n)_{n \geq 1}, (b_n)_{n \geq 1}, a_n \in \Omega(b_n)$ means $\liminf_{n \to \infty} a_n/b_n > 0$. For two sequences of nonnegative random variables $(X_n)_{n \geq 1}, (Y_n)_{n \geq 1}, X_n \in O_{\mathbb{P}}(Y_n)$ means that for any $\epsilon > 0$ there exists $M$ and $n_0$ such

that $\mathbb{P}(X_n/Y_n > M) \leq \epsilon$ for all $n \geq n_0$. The sub-Gaussian norm of a random variable $X$ is $\|X\|_{\psi_2} := \sup_{q \geq 1} q^{-1/2} \mathbb{E}[|X|^q]^{1/q}$, and $X$ is said to be sub-Gaussian if $\|X\|_{\psi_2} < \infty$. This condition can be understood as meaning that the tails of the distribution of $X$ decay at least as quickly as a Gaussian, but includes the case where $X$ is a discrete random variables taking only finitely many different values, or more generally the case where the support of the distribution of $X$ is bounded. The Frobenius and spectral matrix norms are respectively denoted $\|\cdot\|_{\text{F}}$ and $\|\cdot\|$. $\mathbf{I}[\cdot]$ is the indicator function, $\mathbf{I}_p$ is the $p$-by-$p$ identity matrix.

## 2. A generalised Hanson-Wright inequality

The Hanson-Wright (HW) inequality is a concentration inequality for the quadratic form: $\mathbf{X}^\top \mathbf{A} \mathbf{X}$, where $\mathbf{X}$ is a vector of independent, sub-Gaussian random variables and $\mathbf{A}$ is a matrix (Hanson & Wright, 1971; Wright, 1973; Rudelson & Vershynin, 2013), see supplementary material for more background. Our GHW inequality concerns $\mathbf{X}^\top \mathbf{A} \mathbf{X}'$, where $\mathbf{X}, \mathbf{X}'$ are allowed to be dependent, reducing to the standard HW inequality of Rudelson & Vershynin (2013) when $\mathbf{X} = \mathbf{X}'$.

**Theorem 2.1.** *Let* $\mathbf{X} = (X_1, \ldots, X_p)$ *and* $\mathbf{X}' = (X_1', \ldots, X_p')$ *be* $\mathbb{R}^p$*-valued random vectors such that the pairs* $(X_j, X_j'), j = 1, \ldots, p$ *are mutually independent, and* $\mathbb{E}[X_j] = \mathbb{E}[X_j'] = 0$ *and* $\max(\|X_j\|_{\psi_2}, \|X_j'\|_{\psi_2}) \leq K$ *for all* $j = 1, \ldots, p$. *Let* $\mathbf{A} \in \mathbb{R}^{p \times p}$. *Then there is an absolute constant* $c$ *such that for any* $t \geq 0$,

$$\mathbb{P}\left(\left|\mathbf{X}^\top \mathbf{A} \mathbf{X}' - \mathbb{E}[\mathbf{X}^\top \mathbf{A} \mathbf{X}']\right| > t\right)$$
$$\leq 2 \exp\left[-c \min\left\{\frac{t^2}{K^4 \|\mathbf{A}\|_{\text{F}}^2}, \frac{t}{K^2 \|\mathbf{A}\|}\right\}\right].$$

**Zero-mean i.i.d. vectors are close to orthogonal.** As a first illustration of how Theorem 2.1 can be used to understand data geometry, we re-visit a conventional perspective on zero-mean, i.i.d. random vectors. Let $\mathbf{X}_1, \ldots, \mathbf{X}_n$ be i.i.d. copies of the random vector $\mathbf{X}$ in Theorem 2.1 subject to the additional requirement that its elements satisfy $\mathbb{E}[|X_j|^2] = 1$ for $j = 1, \ldots, p$, let $\mathbf{\Sigma} \in \mathbb{R}^{p \times p}$ be positive semidefinite and define $\mathbf{Y}_i := \mathbf{\Sigma}^{1/2} \mathbf{X}_i$. Then $\mathbf{Y}_1, \ldots, \mathbf{Y}_n$ are i.i.d., each have mean zero and covariance matrix $\mathbf{\Sigma}$, and for $i \neq j, \mathbb{E}[\mathbf{Y}_i \cdot \mathbf{Y}_j] = 0$. We define the *effective dimension*:

$$p_{\text{eff}} := \frac{\text{tr}(\mathbf{\Sigma})}{\|\mathbf{\Sigma}\|}. \tag{2}$$

We have $1 \leq p_{\text{eff}} \leq \text{rank}(\mathbf{\Sigma}) \leq p$, with $p_{\text{eff}} = p$ when all $p$ eigenvalues of $\mathbf{\Sigma}$ are equal. Since $\text{tr}(\mathbf{\Sigma}) = \mathbb{E}[\|\mathbf{Y}_i - \mathbb{E}[\mathbf{Y}_i]\|^2]$, $p_{\text{eff}}$ can be understood as a normalised measure of total variability of $\mathbf{Y}_i$. The quantity $\text{tr}(\mathbf{\Sigma})/\|\mathbf{\Sigma}\|$ was termed *intrinsic dimension* by Tropp et al. (2015) and Vershynin (2018). Here, we refer to it as *effective dimension* in order to

avoid confusion with the notion of *latent intrinsic dimension* defined in Section 5.

**Proposition 2.2.** *Let $\mathbf{Y}_1, \ldots, \mathbf{Y}_n$ be the i.i.d., zero-mean random vectors defined above. If $p_{\mathrm{eff}} \in \Omega(\log n)$, then*

$$\max_{i,j \in [n]} \left| \frac{\mathbf{Y}_i \cdot \mathbf{Y}_j}{\mathrm{tr}(\mathbf{\Sigma})} - \mathbf{I}[i = j] \right| \in O_{\mathbb{P}} \left( \sqrt{\frac{\log n}{p_{\mathrm{eff}}}} \right). \quad (3)$$

The proof of proposition 2.2 involves applying Theorem 2.1 with $\mathbf{X} = \mathbf{X}_i$, $\mathbf{X}' = \mathbf{X}_j$, $\mathbf{A} = \mathbf{\Sigma}$, so that $\mathbf{X}^{\top} \mathbf{A} \mathbf{X} = \mathbf{Y}_i \cdot \mathbf{Y}_j$, and taking a union bound over $i, j \in [n]$. Proposition 2.2 tells us that if $p_{\mathrm{eff}} \in \omega(\log n)$, then $\mathbf{Y}_1, \ldots, \mathbf{Y}_n$ tend to be uniformly close to orthogonal, and have magnitudes uniformly close to $\mathrm{tr}(\mathbf{\Sigma})$. This geometric configuration is often referred to as "high-dimensional" behaviour of random vectors, but Proposition 2.2 shows that "high-dimensional" is perhaps a misnomer here, since only $p_{\mathrm{eff}} \in \omega(\log n)$ is required.

# 3. Topology and manifold structure in data emerges from random functions

**Model definition and statistical properties.** In this section we define a general form of random function statistical model which is substantially more expressive than the i.i.d. setting of Proposition 2.2. The setup of the model follows Whiteley et al. (2026) in part but not all; our model allows for more general dependence structure across features. Similarly to Whiteley et al. (2026) though, it is not our aim to perform confirmatory statistical analysis or model fitting, rather the utility of the model is to help us understand the performance of TDA and manifold learning. The main ingredients of the model are:

- a metric space $(\mathcal{Z}, d_{\mathcal{Z}})$, where $\mathcal{Z}$ is a set and $d_{\mathcal{Z}}(\cdot, \cdot)$ is a distance function, and a collection of points $z_1, \ldots, z_n \in \mathcal{Z}$. We write $\mathcal{Z}_n := \{z_1, \ldots, z_n\}$.
- random, $\mathbb{R}$-valued functions, $X_1(\cdot), \ldots, X_p(\cdot)$, each with domain $\mathcal{Z}$, so for each $z \in \mathcal{Z}$ and $j \in [p]$, $X_j(z)$ is a random variable. We write in vector form $\mathbf{X}(z) := (X_1(z), \ldots, X_p(z))$.
- positive semidefinite matrices $\{\mathbf{\Sigma}(z) \in \mathbb{R}^{p \times p}; z \in \mathcal{Z}\}$ and vectors $\{\boldsymbol{\mu}(z) \in \mathbb{R}^p; z \in \mathcal{Z}\}$;
- random 'noise' vectors $\mathbf{E}_1, \ldots, \mathbf{E}_n \in \mathbb{R}^p$, which are independent of the $X_j$.

We then define, with $\sigma \geq 0$,

$$\mathbf{Y}_i := \mathbf{\Sigma}^{1/2}(z_i) \mathbf{X}(z_i) + \boldsymbol{\mu}(z_i) + \sigma \mathbf{E}_i.$$

We denote the 'noise-free' component of the model $\mathbf{Y}^{\mathrm{nf}}(z) := \mathbf{\Sigma}^{1/2}(z) \mathbf{X}(z) + \boldsymbol{\mu}(z)$, so that $\mathbf{Y}_i \equiv \mathbf{Y}^{\mathrm{nf}}(z_i) + \sigma \mathbf{E}_i$.

**A 1.** The random functions $X_j$ are independent across $j$, and for every $j$ and $z$, $\mathbb{E}[X_j(z)] = 0$, $\mathbb{E}[|X_j(z)|^2] = 1$ and $\|X_j(z)\|_{\psi_2} \leq K$, for some $K < \infty$.

**A 2.** The random vectors $\mathbf{E}_1, \ldots, \mathbf{E}_n$ are i.i.d., $\mathbb{E}[\mathbf{E}_i] = \mathbf{0}$, $\mathbb{E}[\mathbf{E}_i \mathbf{E}_i^{\top}] = \mathbf{I}_p$, and the elements $E_i^{(j)}$, $j = 1, \ldots, p$ of each $\mathbf{E}_i$ are independent and for all $j$, $\|E_i^{(j)}\|_{\psi_2} < K$ for some $K < \infty$.

This model relaxes both the *independence* and *identical distribution* parts of the i.i.d. assumption in Proposition 2.2: we allow that for any $j \in [p]$ and $z, z' \in \mathcal{Z}$, $X_j(z)$ and $X_j(z')$ may be dependent, hence $\mathbf{Y}_i$ and $\mathbf{Y}_j$ may be dependent; and **A1** and **A2** imply that $\mathbf{Y}_i$ is a random vector with mean $\boldsymbol{\mu}(z_i)$ and covariance matrix $\mathbf{\Sigma}(z_i) + \sigma^2 \mathbf{I}_p$. Theorem 3.1 shows that despite this general dependence structure, dot products amongst vectors $\mathbf{Y}_1, \ldots, \mathbf{Y}_n$ are concentrated about their expectations in a manner analogous to Proposition 2.2. Extending the notion of effective dimension to the present setting, we define:

$$p_{\mathrm{eff}}^{(i)} := \frac{\mathrm{tr}[\mathbf{\Sigma}(z_i)]}{\|\mathbf{\Sigma}(z_i)\|}.$$

**Theorem 3.1.** *Assume that $\mathbf{Y}_1, \ldots, \mathbf{Y}_n$ follow the random function model, **A1**-**A2** hold, and $\min_{i \in [n]} p_{\mathrm{eff}}^{(i)} \in \Omega(\log n)$ as $n \to \infty$. Then*

$$\max_{i,j \in [n]} \left| \frac{\mathbf{Y}_i \cdot \mathbf{Y}_j}{\mathbb{E}[\|\mathbf{Y}_i\|^2]^{\frac{1}{2}} \mathbb{E}[\|\mathbf{Y}_j\|^2]^{\frac{1}{2}}} - \frac{\mathbb{E}[\mathbf{Y}_i \cdot \mathbf{Y}_j]}{\mathbb{E}[\|\mathbf{Y}_i\|^2]^{\frac{1}{2}} \mathbb{E}[\|\mathbf{Y}_j\|^2]^{\frac{1}{2}}} \right|$$
$$\in O_{\mathbb{P}} \left( \sqrt{\frac{\log n}{\min_{i \in [n]} p_{\mathrm{eff}}^{(i)}}} \right),$$

*as $n \to \infty$.*

We shall next see that under additional but mild assumptions, the quantities $\mathbb{E}[\|\mathbf{Y}_i\|^2]$ and $\mathbb{E}[\mathbf{Y}_i \cdot \mathbf{Y}_j]$ have a rich geometric interpretation, which can be transferred to $\mathcal{Y}_n = \{\mathbf{Y}_1, \ldots, \mathbf{Y}_n\}$ by Theorem 3.1.

**Relationship between point-clouds $\mathcal{Y}_n$ and $\mathcal{M}_n$.** Consider the kernel function:

$$(z, z') \mapsto \mathbb{E}[\mathbf{Y}^{\mathrm{nf}}(z) \cdot \mathbf{Y}^{\mathrm{nf}}(z')]. \quad (4)$$

and the assumptions:

**A3.** $z \mapsto \mathbf{Y}^{\mathrm{nf}}(z)$ is mean-square continuous, i.e., $z \to z'$ implies $\mathbb{E}[\|\mathbf{Y}^{\mathrm{nf}}(z) - \mathbf{Y}^{\mathrm{nf}}(z')\|^2] \to 0$.

**A4.** The metric space $(\mathcal{Z}, d_{\mathcal{Z}})$ is compact.

Assumption **A3** distinguishes each dimension of $\mathbf{Y}^{\mathrm{nf}}(z_i)$ as "signal" rather than noise — cf. $\mathbf{E}_i$ being independent across $i$ under **A2** — ensuring the informative signal grows with $p$. Moreover, under **A3** the kernel function in (4) is

continuous. If, in addition, **A4** holds, for any finite Borel measure $\nu$ supported on $\mathcal{Z}$, Mercer's theorem (Steinwart & Christmann, 2008)[Thm 4.49] tells us that the kernel function in (4) has the representation:

$$\mathbb{E}[\mathbf{Y}^{\mathrm{nf}}(z) \cdot \mathbf{Y}^{\mathrm{nf}}(z')] = \phi(z) \cdot \phi(z') = \sum_{k=1}^{r} \lambda_k u_k(z) u_k(z')$$
(5)

where $\phi : \mathcal{Z} \to \ell_2$ is conventionally called a *feature map* and the $k$th element of the vector $\phi(z)$ is $\lambda_k^{1/2} u_k(z)$, where the $u_k$ are $L_2(\nu)$-orthonormal functions, the $\lambda_k$ are associated eigenvalues and $r \in \{1, 2, \ldots\} \cup \{\infty\}$ is the largest $k$ such that $\lambda_k > 0$. We call $r$ *correlation rank*, which can be understood as a measure of functional complexity of the kernel function, and hence the correlation between $\mathbf{Y}^{\mathrm{nf}}(z)$ and $\mathbf{Y}^{\mathrm{nf}}(z')$: $r$ counts the number of orthonormal functions needed to express the dot product $\phi(z) \cdot \phi(z')$ across all $z, z' \in \mathcal{Z}$. Define:

$$\mathcal{M} := \{\phi(z);\, z \in \mathcal{Z}\}, \qquad \mathcal{M}_n := \{\phi(z_i);\, i \in [n]\}.$$

It follows from (5) and **A2** that

$$\mathbb{E}[\mathbf{Y}_i \cdot \mathbf{Y}_j] = \phi(z_i) \cdot \phi(z_j) + p\sigma^2 \mathbf{I}[i = j].$$
(6)

Substituting (6) into Theorem 3.1, we see that when $\min_i p_{\mathrm{eff}}^{(i)} \gg \log n$, with high probability, $\mathbf{Y}_i \cdot \mathbf{Y}_j \approx \phi(z_i) \cdot \phi(z_j) + p\sigma^2 \mathbf{I}[i = j]$, implying that the geometry of the point-cloud $\mathcal{Y}_n$ resembles that of $\mathcal{M}_n$ up to some distortion depending on $\sigma$. The normalizing terms $\mathbb{E}[\|\mathbf{Y}_i\|^2]$ appearing in Theorem 3.1 are typically unknown in practice; Proposition 3.2 explains the behaviour of $\mathbf{Y}_i \cdot \mathbf{Y}_j$ and $\mathsf{CosSim}(\mathbf{Y}_i, \mathbf{Y}_j)$ avoiding unknown normalization.

**Proposition 3.2.** *If **A1**-**A4** hold, then:*

$$\max_{i,j \in [n]} \left| \frac{\mathbf{Y}_i \cdot \mathbf{Y}_j}{p} - \frac{\phi(z_i) \cdot \phi(z_j)}{p} - \sigma^2 \mathbf{I}[i = j] \right| \in$$

$$O_{\mathbb{P}}\left( \left[ \max_{i \in [n]} \frac{\mathrm{tr}[\mathbf{\Sigma}(z_i)] + \|\boldsymbol{\mu}(z_i)\|^2}{p} + \sigma^2 \right] \sqrt{\frac{\log n}{\min_{i \in [n]} p_{\mathrm{eff}}^{(i)}}} \right)$$

*and with*

$$\gamma_{ij}(\sigma) := \gamma_i(\sigma)\gamma_j(\sigma), \quad \gamma_i(\sigma) := \frac{(\|\phi(z_i)\|^2 + p\sigma^2)^{1/2}}{\|\phi(z_i)\|},$$

$$\max_{i \neq j \in [n]} \left| \mathsf{CosSim}(\mathbf{Y}_i, \mathbf{Y}_j) - \frac{\mathsf{CosSim}(\phi(z_i), \phi(z_j))}{\gamma_{ij}(\sigma)} \right|$$

$$\in O_{\mathbb{P}}\left( \sqrt{\frac{\log n}{\min_{i \in [n]} p_{\mathrm{eff}}^{(i)}}} \right).$$

Recalling (1), the first part of Proposition 3.2 implies that if $\max_{i \in [n]} /(\mathrm{tr}[\mathbf{\Sigma}(z_i)] + \|\boldsymbol{\mu}(z_i)\|^2)/p + \sigma^2 \in O(1)$ and

$\min_i p_{\mathrm{eff}}^{(i)} \in \omega(\log n)$, then for $i \neq j$,

$$\frac{1}{p}\|\mathbf{Y}_i - \mathbf{Y}_j\|^2 \approx \frac{1}{p}\|\phi(z_i) - \phi(z_j)\|^2 + 2\sigma^2.$$
(7)

For the second part of Proposition 3.2, notice that $\gamma_{ij}(0) = 1$, and for any $\sigma > 0$, if $\|\phi(z)\|$ is constant in $z$ (e.g. $\mathcal{Z}$ is a vector space and the kernel (4) is a function of $z - z'$), then $\gamma_{ij}^{(\sigma)}$ is constant in $i, j$.

**Relationship between metric space $\mathcal{Z}$ and manifold $\mathcal{M}$.** Lemma 3.3 shows that under a mild non-degeneracy condition on $\mathbf{Y}^{\mathrm{nf}}(\cdot)$, $\mathcal{M}$ is topologically equivalent to $\mathcal{Z}$.

**A5.** If $z \neq z'$, then $\mathbb{E}[\|\mathbf{Y}^{\mathrm{nf}}(z) - \mathbf{Y}^{\mathrm{nf}}(z')\|^2] > 0$.

**Lemma 3.3.** *Under **A3**-**A5**, $\phi$ is a homeomorphism between $\mathcal{Z}$ and $\mathcal{M}$, i.e., $\phi$ is continuous, invertible on its image, and has a continuous inverse.*

Informally, this means $\mathcal{M}$ resembles $\mathcal{Z}$ subject to some transformation such as bending, twisting or stretching, but not cutting, or puncturing; $\mathcal{Z}$ and $\mathcal{M}$ must have the same number of connected components, the same number of one-dimensional holes, two-dimensional cavities, and so on. The term *topological manifold* conventionally refers to some topological space which is *locally* homeomorphic to a subset of Euclidean space; we can therefore speak of $\mathcal{M}$ as a topological manifold, but one which is *globally* homeomorphic to the metric space $\mathcal{Z}$.

In combination, Theorem 3.2, Proposition 3.1 and Lemma 3.3 tell us that if the points in $\mathcal{Z}_n$ are distributed across $\mathcal{Z}$, then $\mathcal{M}_n$ and hence $\mathcal{Y}_n$ will convey the 'shape' of $\mathcal{Z}$ – in the remaining sections of the paper we will explore implications of this in TDA and manifold learning.

# 4. Consistency of persistence diagrams

In this section we discuss how, under the model from Section 3, TDA can be applied to $\mathcal{Y}_n$ in order to estimate certain topological characteristics of $\mathcal{M}$ and hence $\mathcal{Z}$. In the mathematical framework of persistent homology, a topological space has associated *Betti numbers* $H_0, H_1, H_2, \ldots$, respectively indicating the number of connected components, 1D holes, 2D cavities, etc., which the space exhibits. Two homeomorphic spaces, such as $\mathcal{Z}$ and $\mathcal{M}$ in the setting of Lemma 3.3, have the same Betti numbers. TDA techniques (Edelsbrunner et al., 2002; Zomorodian & Carlsson, 2004; Chazal et al., 2009; Chazal & Michel, 2021), for example implemented in the python package `ripser` (Tralie et al., 2018), allow *persistence diagrams* associated with data point-clouds to be computed (an example is shown in Figure 3(c)), in turn enabling Betti numbers of the underlying space to be estimated.

We view the sets $p^{-1/2}\mathcal{Y}_n$, $p^{-1/2}\mathcal{M}_n$ and $p^{-1/2}\mathcal{M}$ as metric spaces by equipping them with Euclidean (i.e., $\ell_2$) dis-

tance. We denote by $\mathrm{dgm}(\cdot)$ persistence diagrams under some common choice of filtration[1]. In the supplementary material we also discuss TDA operating on the normalised vectors $\mathbf{Y}_i/\|\mathbf{Y}_i\|$. A careful combination of existing results (Chazal et al., 2013; Ivanov et al., 2016) detailed in the supplementary material gives

$$
\begin{aligned}
d_{\mathrm{b}} & \left( \mathrm{dgm}(p^{-1/2}\mathcal{Y}_n), \mathrm{dgm}(p^{-1/2}\mathcal{M}) \right) \\
& \leq 2(d_{\mathrm{H}}(p^{-1/2}\mathcal{M}_n, p^{-1/2}\mathcal{M}) \\
& \qquad + d_{\mathrm{GH}}(p^{-1/2}\mathcal{Y}_n, p^{-1/2}\mathcal{M}_n)), \quad (8)
\end{aligned}
$$

where $d_{\mathrm{b}}$ is the bottleneck distance, a distance between persistence diagrams; $d_{\mathrm{GH}}$ is the Gromov-Hausdorff distance, a distance between metric spaces; and $d_{\mathrm{H}}$ is the Hausdorff distance, a distance between subsets of a metric space (Burago et al., 2001).

The persistence diagram $\mathrm{dgm}(p^{-1/2}\mathcal{Y}_n)$ is said to be consistent if the left-hand-side of (8) converges to zero in probability. The first term on the right-hand-side can be shown to vanish as $n \to \infty$, if $\phi(z_1), \ldots, \phi(z_n)$ are i.i.d. from a measure on $\mathcal{M}$ satisfying standard regularity conditions (Chazal et al., 2013). Proposition 3.2 allows us to show that, up to an additive term depending only on $\sigma$, *the second term vanishes*, as long as $\max_{i \in [n]}(\mathrm{tr}[\boldsymbol{\Sigma}(z_i)] + \|\boldsymbol{\mu}(z_i)\|^2)/p + \sigma^2 \in O(1)$ and $\min_{i \in [n]} p_{\mathrm{eff}}^{(i)}/\log n \to \infty$:

$$
\begin{aligned}
d_{\mathrm{GH}}^2 & (p^{-1/2}\mathcal{Y}_n, p^{-1/2}\mathcal{M}_n) \\
& \leq \max_{i,j \in [n]} \frac{1}{p} |\mathbf{Y}_i \cdot \mathbf{Y}_j - \phi(z_i) \cdot \phi(z_j)| \\
& \leq \sigma^2 + O_{\mathbb{P}} \left( \sqrt{\frac{\log n}{\min_{i \in [n]} p_{\mathrm{eff}}^{(i)}}} \right),
\end{aligned}
$$

where the first inequality is shown in the supplementary material. This tells us that accurate estimation of topology is possible without requiring $p \gg n$.

This perspective complements recent work describing a *curse of dimensionality* for persistent homology (Damrich et al., 2024; Hiraoka et al., 2024). While those studies demonstrate that accumulating uninformative noise can destabilise persistence diagrams, our model captures scenarios where a growing effective dimension $p_{\mathrm{eff}}$ brings additional, structured signal (via mean-square continuity — assumption **A3**). Thus, while unstructured noise obscures topology, high dimensionality acts as a blessing when new features expand the signal, driving the geometric concentration required for faithful recovery.

[1] which could be either the Rips, Čech, or Alpha filtration.

## 5. Looking for evidence of isometry

Isometry is a stronger form of relationship between $\mathcal{Z}$ and $\mathcal{M}$ than homeomorphism, requiring that shortest path lengths on $\mathcal{M}$ faithfully represent those on $\mathcal{Z}$ (Burago et al., 2001). To define isometry mathematically, consider the random function model with, for some $d \leq \tilde{d}$, $\mathcal{Z}$ a smooth, locally $d$-dimensional subset of $\mathbb{R}^{\tilde{d}}$, with $d_{\mathcal{Z}}$ being Euclidean distance. Examples of such $\mathcal{Z}$ include, in the case of $d = 1$, a line segment, curve or circle (as in the toy example above), and in the case $d = 2$, a disk, sphere, or torus. We refer to $d$ as *latent intrinsic dimension*.

For any two points $z, z' \in \mathcal{Z}$, a *path* with end-points $z, z'$ is a smooth curve $\eta : [0, 1] \to \mathcal{Z}$, such that $\eta_0 = z$ and $\eta_1 = z'$; similarly a path between $x, x' \in \mathcal{M}$ is a smooth curve $\gamma : [0, 1] \to \mathcal{M}$, with $\gamma_0 = x, \gamma_1 = x'$. The lengths of these paths are $L(\eta) := \int_0^1 \|\dot{\eta}_t\| \mathrm{dt}$ and $L(\gamma) := \int_0^1 \|\dot{\gamma}_t\| \mathrm{dt}$. We say that *isometry* holds if for any path $\eta$ in $\mathcal{Z}$ and the path in $\gamma$ defined by $\gamma_t := \phi(\eta_t)$, $L(\gamma)$ and $L(\eta)$ are equal up to a constant of proportionality. In particular this implies that the length of any shortest path in $\mathcal{Z}$ with end-points $z, z'$ is equal to the length of any shortest path in $\mathcal{M}$ with end-points $\phi(z), \phi(z')$. It is known that isometry holds under various sufficient conditions on the kernel function $(z, z') \mapsto \mathbb{E}[\mathbf{Y}^{\mathrm{nf}}(z) \cdot \mathbf{Y}^{\mathrm{nf}}(z')]$ (Whiteley et al., 2021; 2026). However, in practice this kernel function will often be unknown. We can nevertheless assess evidence of isometry given $\mathcal{Y}_n$ and $\mathcal{Z}_n$ in a manner inspired by the manifold learning literature, specifically the first step of Isomap (Tenenbaum et al., 2000).

We approximate shortest path-lengths in $\mathcal{Z}$ by constructing a $k$-nearest[2] neighbour graph with vertex set $\mathcal{Z}_n$ and edge-lengths $\|z_i - z_j\|$ if $z_i$ and $z_j$ are neighbours; then compute shortest paths in this edge-weighted graph. To approximate shortest path-lengths in $\mathcal{M}$, we similarly construct a $k$-nearest neighbour graph with vertex set $\mathcal{Y}_n$ and edge lengths $\|\mathbf{Y}_i - \mathbf{Y}_j\|$. We write $\widehat{L}(z_i, z_j)$ and $\widehat{L}(\mathbf{Y}_i, \mathbf{Y}_j)$ for the resulting shortest path-lengths, noting that these lengths are of the form $\sum_m \|z_{k_m} - z_{k_{m-1}}\|$ and $\sum_m \|\mathbf{Y}_{\ell_m} - \mathbf{Y}_{\ell_{m-1}}\|$ for some $[n]$-valued sequences $(k_m)$ and $(\ell_m)$. If isometry holds with constant of proportionality $\beta$, then when $z$ is close to $z'$, $\|\phi(z) - \phi(z')\| \approx \beta \|z - z'\|$. This suggests assessing evidence of isometry through linear regression of the values $(\widehat{L}(\mathbf{Y}_i, \mathbf{Y}_j); i \neq j)$ on to the associated values $(\widehat{L}(z_i, z_j); i \neq j)$ allowing for a positive intercept to account for noise (informed by (7)). The correlation coefficient, $\rho$, associated with this linear regression numerically quantifies the presence of isometry, with a maximum value of $\rho = 1$ corresponding to a perfect linear fit.

[2] a default choice of $k$ is to choose it to be the smallest value such that the graph is connected

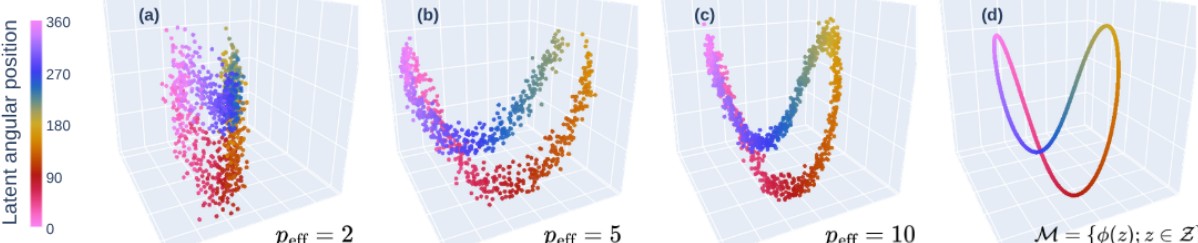

*Figure 1.* Ambient intrinsic dimension $p_{\text{eff}}$, correlation rank $r$, and latent intrinsic dimension $d$ at play in simulation from a toy example of the random function model with $n = 1000$. (a)–(c) show SVD visualisation of simulated data with respectively $p_{\text{eff}} = 2, 5, 10$; as $p_{\text{eff}}$ grows, the $d = 1$-dimensional manifold $\mathcal{M} = \{\phi(z); z \in \mathcal{Z}\}$ shown in (d) emerges in a $r = 3$-dimensional subspace. In this example $\mathcal{M}$ is homeomorphic to the latent space $\mathcal{Z}$, which is a circle. See Section 3 for details.

# 6. Experiments

## 6.1. Toy example

Let $\mathcal{Z}$ be a circle, with

$$\mathcal{Z} = \{z = (z_1, z_2) : z_1^2 + z_2^2 = 1\}, \quad r = 3,$$

$$\phi(z) = p^{1/2}[z_1, \frac{2}{\pi}\sin(\pi z_2/2), \frac{2}{\pi}\cos(\pi z_2/2)].$$

The random functions $X_j$ are i.i.d., zero-mean Gaussian processes each with covariance function $p^{-1}\phi(z) \cdot \phi(z')$, $\boldsymbol{\mu}(z) = 0$, $\boldsymbol{\Sigma}(z) = \mathbf{I}_p + \frac{1}{p}\mathbf{11}^\top$ and the elements of $\mathbf{E}_i$ are distributed $\mathcal{N}(0, 1)$, and $\sigma^2 = 0.02$. Here, $p_{\text{eff}}^{(i)} = (p+1)/2$. Figure 1 shows the first $r = 3$ dimensions of the SVD embedding of $\mathcal{Y}_n$ for $p_{\text{eff}}^{(i)} = 2, 5, 10$, with $n = 1000$, and $z_1, \ldots, z_{1000}$ uniformly spaced around $\mathcal{Z}$. As $p_{\text{eff}}^{(i)}$ increases the manifold $\mathcal{M}$ emerges, and is homeomorphic to $\mathcal{Z}$.

## 6.2. Empirical estimates of $p_{\text{eff}}$

Before turning to a detailed neuroscience case study on grid cell activity, we estimate the effective dimension $p_{\text{eff}}$ on a small collection of benchmark datasets spanning biological, textual, image, and time series modalities, including the grid cell activity. Full dataset descriptions are provided in Appendix J.

For each dataset, we estimate $p_{\text{eff}}$ using $\widehat{p_{\text{eff}}} := \text{tr}(\widehat{\boldsymbol{\Sigma}})/\|\widehat{\boldsymbol{\Sigma}}\|$, where $\widehat{\boldsymbol{\Sigma}}$ denotes the sample covariance matrix of the centred data. In Appendix I, we investigate the effect of finite sample size on the estimator $\widehat{p_{\text{eff}}}$, showing that it is systematically biased downwards, but asymptotically converges to the true value. Thus, the estimator represents a conservative lower bound. Table 1 reports the resulting effective dimension estimates, along with the ambient dimension $p$ and sample size $n$ for each dataset. Several datasets, including the grid cell activity, exhibit effective dimensions in the order of $\log n$. This places them in a *moderate* effective-dimension regime, where our theory predicts that observed distances should reflect the geometry of an underlying latent manifold, but with non-negligible noise.

| Dataset | $p$ | $n$ | $\widehat{p_{\text{eff}}}$ | $\log n$ |
|---|---|---|---|---|
| Grid cell activity | 149 | 15,000 | 12.67 | 9.62 |
| Newsgroups | 12,864 | 11,314 | 163.41 | 9.33 |
| Planaria (stem cells) | 5,821 | 8,959 | 37.55 | 9.10 |
| Amazon Reviews | 5,588 | 5,000 | 141.75 | 8.52 |
| MNIST | 784 | 5,000 | 10.21 | 8.52 |
| S&P 500 | 1,258 | 470 | 12.13 | 6.15 |

*Table 1.* Effective dimension estimates $\widehat{p_{\text{eff}}}$ for representative machine learning datasets.

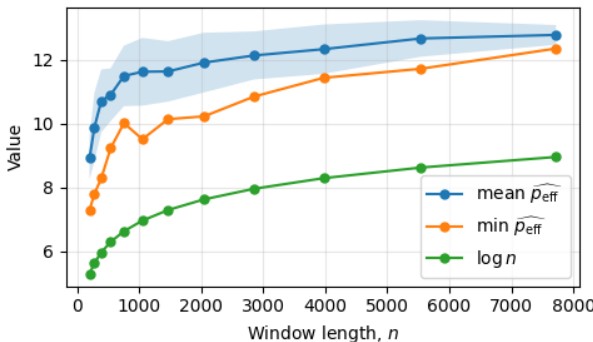

*Figure 2.* $\widehat{p_{\text{eff}}}$ for the grid cell data as a function of window length $n$. Curves show the mean $\pm 1$ s.d. and the minimum across 30 contiguous time windows.

While the estimator $\widehat{p_{\text{eff}}}$ uses the sample covariance and reflects effective dimension under the assumption that $\boldsymbol{\Sigma}(z)$ is constant, $p_{\text{eff}}$ may vary across the latent space and in many applications $n$ may not be as large. To explore this, Figure 2 shows the minimum and mean $\widehat{p_{\text{eff}}}$ over 30 contiguous time windows for the grid cell data, along with $\log n$ for comparison. Note that for this example, a short time window corresponds to data being confined to a small region of physical space, which we posit in the following section to be the underlying latent space. The quantity $\min \widehat{p_{\text{eff}}}$ corresponds most closely to the quantity in the asymptotic rates of Theorem 3.1 and Proposition 3.2.

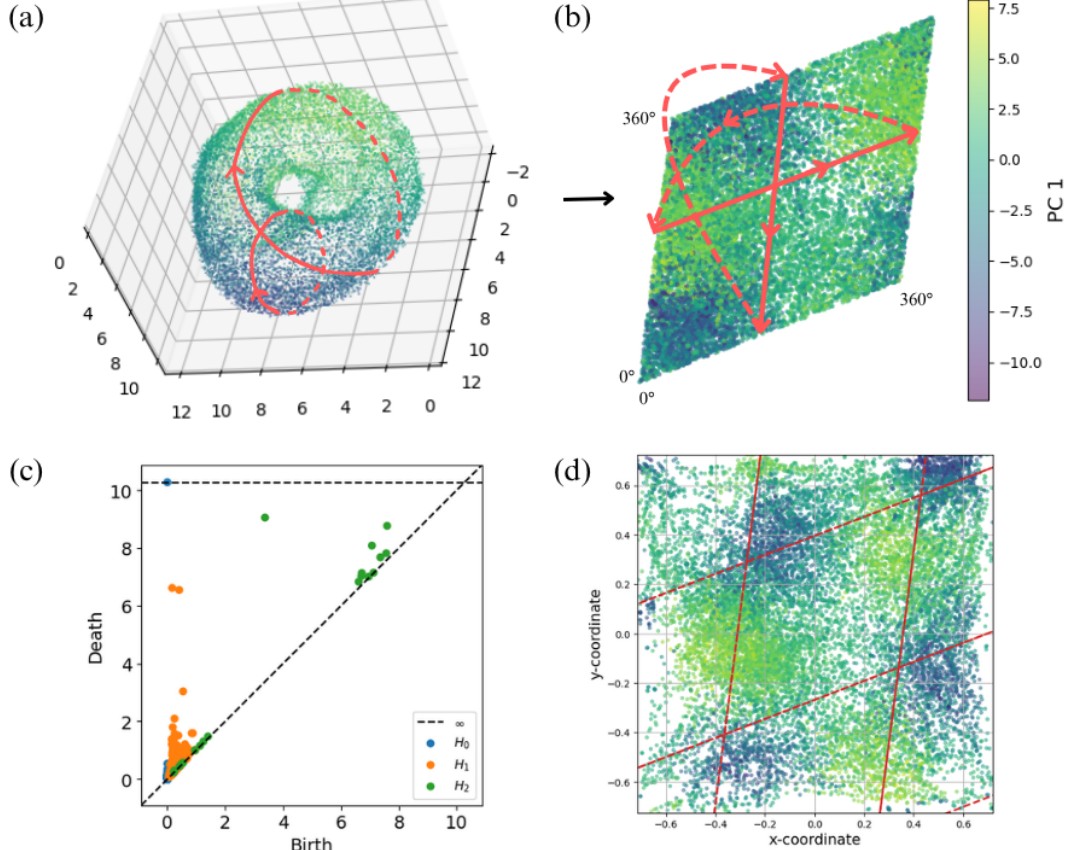

*Figure 3.* Re-creating the grid cell analysis of Gardner et al. (2022). (a): UMAP visualisation of the grid cell data $\mathbf{Y}_1, \ldots, \mathbf{Y}_n$. This visualisation suggest the presence of toroidal structure, confirmed by the persistence diagram in (c), indicating Betti numbers $H_0 = 1$, $H_1 = 2$, $H_2 = 1$. In (b), cohomological decoding maps the circular coordinates of the torus to a rhombus. (d) shows how these coordinates correspond to physical space through tesselation of the rhombus. In (a), (c) and (d), points are colored by the first component in the PCA embedding of the data to aid visual recognition of the torus.

### 6.3. Grid cell activity

Understanding how spatial location is represented in brain activity is a fundamental neuroscience challenge. In 2014, May-Britt Moser and Edvard I. Moser received a Nobel prize for their discovery of *grid cells* – nerve cells whose firing activity is associated with a grid of spatial locations (Fyhn et al., 2004; Hafting et al., 2005). Combined with earlier work of co-prize-winner John O'Keefe (1976), this discovery showed that the brain effectively creates a map of the world around us through the firing patterns of neurons. In their 2022 *Nature* paper, Gardner et al. (2022) established a deeper understanding of grid cell activity using persistent homology techniques, revealing the presence of toroidal structure. We focus on a subset of the experiments they reported, in which a rat was confined to a square enclosure ("open field"), and the firing of $p \approx 150$ grid cells was co-recorded and sub-sampled into $n = 15,000$ time bins of length 10ms, while simultaneously recording the rats physical position in the enclosure, which we denote $\xi_1, \ldots, \xi_n$. Representing the neural activity data as $\mathbf{Y}_1, \ldots, \mathbf{Y}_n$, the

$j$th element of $\mathbf{Y}_i$ is the firing rate of the $j$th grid cell in the $i$th time bin. After pre-processing $\mathbf{Y}_1, \ldots, \mathbf{Y}_n$ (details in supplementary material), Gardner et al. (2022) made the following two key findings:

1. Persistent homology analysis of $\mathbf{Y}_1, \ldots, \mathbf{Y}_n$ gave estimated Betti numbers of $H_0 = 1$ (one connected component), $H_1 = 2$ (two 1D 'holes' ) and $H_2 = 1$ (one 2D cavity), indicating presence of a torus.
2. Circular coordinates corresponding to the two 1D holes in the torus were found to be associated with physical locations through a two-step transformation: firstly, "unwrapping" the surface of the torus to form a rhombus (moving along one edge of the rhombus corresponds to moving $0 - 360°$ degrees around one of the 1D holds in the torus); and secondly, covering the physical enclosure with a tessellation of this rhombus.

Using the data and code of Gardner et al. (2022), available at Gardner et al. (2021b;a), we re-created this analysis; implementation details are provided in Appendix H. The results are shown in Figure 3.

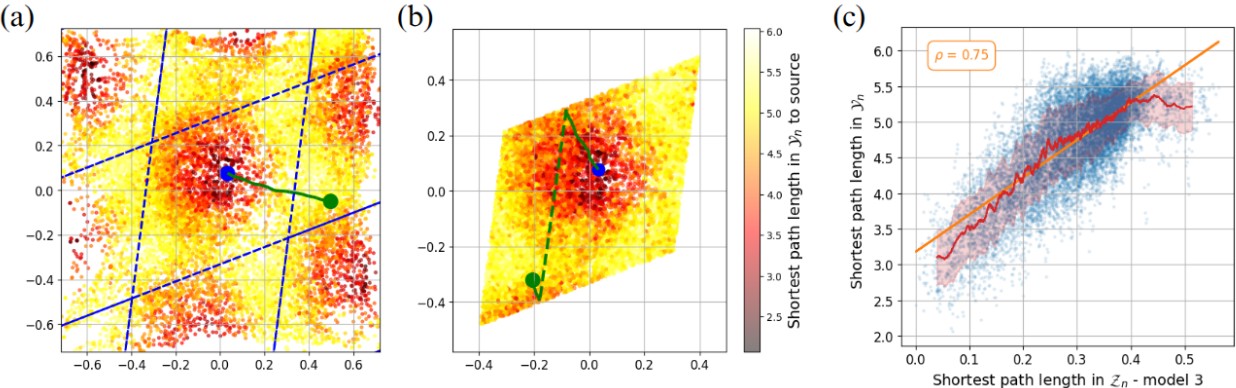

*Figure 4.* **(a)** The tessellated rhombus is plotted atop the physical locations. Two physical locations (source: blue, sink: green), and the shortest path between them in physical space is shown. In (a)-(b), points are colored by the shortest path-length in $\mathcal{Y}_n$ from the source. **(b)** The flat torus model is represented by superimposing the physical locations onto the central rhombus in the tessellation in (a). The shortest path to the same sink point is shown, where distances on the flat torus allow for 'teleporting' (dashed line) at the boundaries. **(c)** Relationship between shortest path-lengths in $\mathcal{Y}_n$ and in $\mathcal{Z}_n$. The orange line shows best linear fit and the red line shows a moving average (with shading for $\pm 1$s.d.).

**Grid cell activity as a random function model.** We interpret the grid cell experiment through the framework of the random function model. Using the cohomological decoding analysis of Gardner et al. (2022), we extract the rhomboidal vectors defining the flat torus model of physical space. Figures 3(a)-(b) illustrate how the flat torus corresponds to the UMAP visualisation of the data, which resembles a torus. Figures 4(a)-(b) illustrate how physical locations $\xi_1, \ldots, \xi_n$ relate to locations $z_1, \ldots, z_n$ on the flat torus, which we denote $\mathcal{Z}$. Opposite edges of this rhombus are identified, allowing paths on $\mathcal{Z}_n$ to 'teleport' at the edges to the corresponding point on the opposite edge.

We view the observed neural activity as generated according to

$$\mathbf{Y}_i := \mathbf{\Sigma}^{1/2}(z_i)\mathbf{X}(z_i) + \boldsymbol{\mu}(z_i) + \sigma\mathbf{E}_i,$$

where $\mathbf{X}_j(z)$ represents the firing-rate modulation of neuron $j$ as a function of position, $\boldsymbol{\mu}(z)$ is the mean firing-rate map, $\mathbf{\Sigma}(z)$ captures position-dependent covariance across neurons, and $\mathbf{E}_i$ represents independent noise.

**From toroidal topology to geometric faithfulness.** As shown in Gardner et al. (2022) and re-created in Figure 3(c), persistent homology of the point cloud $\{\mathbf{Y}_1, \ldots, \mathbf{Y}_n\}$ reveals toroidal topology. This observation is explained by the results of Section 4: when the effective dimension is sufficiently large, persistence diagrams constructed from the neural data recover the topology of the latent manifold.

Taking this a step further, our theory allows us to investigate a stronger geometric relationship. If grid cell population activity satisfies the assumptions of our model, (7) explains why neural distances faithfully reflect the intrinsic neural manifold (a torus), while manifold-learning methods are needed to relate this manifold back to physical space. To

assess this empirically, we follow the approach of Section 5 and approximate geodesic distances on $\mathcal{M}$ using $k$-nearest-neighbour graphs constructed on $\mathcal{Y}_n$. In Figure 4(c), we compare these distances with geodesic distances from a fixed source point on the flat torus model $\mathcal{Z}_n$; an example path is illustrated in Figure 4(b). As per Section 5, an isometric relationship between $\mathcal{Z}$ and $\mathcal{M}$ would manifest itself in a linear relation between shortest path-lengths, with a positive intercept due to noise. We show a straight line fit from OLS in orange and report the corresponding correlation coefficient, $\rho = 0.75$. The red line shows a moving average (with window size $0.01n$), and shading indicates $\pm 1$s.d. The near-linear relationship provides strong empirical evidence of global isometry between the neural manifold $\mathcal{M}$ and the flat torus representation of physical space. Consistent with the moderate effective-dimension regime identified in Section 6.2, this relationship is accompanied by substantial scatter around the linear fit, as expected under non-negligible noise. The deviation from linearity at larger distances is likely attributable to estimation error in the cohomological decoding of the rhombus vectors, which affects the flat torus metric primarily at longer path lengths.

**Relation to prior work on conformal isometry.** Several recent works have investigated the conformal isometry hypothesis that grid cells form locally distance-preserving representation of space. Xu et al. (2024) and Schøyen et al. (2025) analyse how grid cell firing patterns should behave under the assumption of conformal isometry, and show that the resulting predictions are consistent with the hexagonal structure observed in experimental data.

These works take conformal isometry as a modelling assumption and study its consequences. In contrast, our con-

tribution is complementary: we provide a statistical explanation for when and why such geometric structure can be recovered from noisy, finite-dimensional neural data. In particular, our results show that the classical $p \gg n$ regime is not required; rather, effective dimension $p_{\text{eff}}$ sufficiently large compared to $\log n$ ensures that neural distances concentrate around their latent counterparts. In this regime, we find empirical evidence for global isometry between the neural manifold and a flat torus model of physical space.

## 7. Discussion

The generic structure of our statistical model does not constrain us to a specific parametric family of distributions. However, our $\sqrt{\log n / p_{\text{eff}}}$ convergence rates are directly linked to the sub-Gaussian assumptions we make. Weakening to sub-exponential or polynomial moment conditions would result in slower convergence rates, $\log n$ being replaced by some faster-growing function of $n$, connecting our results back to the HDLSS setting of Hall et al. (2005) under their very mild fourth-moment assumptions. We provide an empirical demonstration of this degradation in Appendix B. If stronger distributional assumptions were adopted (e.g., Gaussian, as in Gaussian process latent variable models (Lawrence, 2003; 2005)), one could pursue likelihood-based inference and formal model comparison. We view such parametric approaches as complementary to, but distinct from, the non-parametric and geometric perspective adopted in this work.

More broadly, our results suggest that the limiting "close-to-orthogonal" geometry of high-dimensional data, as characterised in Proposition 2.2, represents only one possible asymptotic regime. Theorem 3.1 and Proposition 3.2 instead imply a wide range of alternative geometric behaviours, governed by the effective dimension $p_{\text{eff}}$. The 'thin-shell' concentration phenomena and near-orthogonality of high-dimensional data have entered into machine learning folklore and are often relied upon when arguing that learning methods suffer from the *curse of dimensionality* (e.g., Hastie et al. (2009)). The framework developed here provides a setting in which such arguments may be revisited, and in which meaningful latent geometric structure can persist even when $p \gg n$ is not satisfied.

## Acknowledgements

H.S. acknowledges support from the EPSRC Centre for Doctoral Training in Computational Statistics and Data Science: COMPASS (EP/S023569/1).

## Impact Statement

This paper presents theoretical work whose main goal is to advance the mathematical understanding of high-dimensional data geometry in machine learning. We do not anticipate any direct ethical or societal impacts arising from this work. While the results may inform the use of manifold learning and representation learning methods across application domains, we believe any broader societal consequences are indirect and align with those commonly associated with advances in the theoretical foundations of machine learning.

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

# Appendices

## A. Related Work

**Curse versus blessing of dimension.** The phrase "*curse of dimensionality*" is often used to convey the general idea that a large number of features, $p$, can be statistically or computationally problematic for a variety of learning methods. Amongst several manifestations of this issue, Hastie et al. (2009, Sec 2.5) highlight that $n$ i.i.d. samples from the uniform distribution on the mean-zero $p$-dimensional unit ball tend to all like close to its surface as $p \to \infty$ grows. Refined analysis of this phenomenon is developed in Cai et al. (2013). Our work contrasts with this perspective: in the setting of the random function model we introduce, $p_{\text{eff}} \to \infty$ leads to the emergence of latent topology and manifold structure in data, arguably more of a *blessing of dimensionality* (Kainen[1], 1997; Donoho, 2000). See Lawrence (2011, Sec. 2.1) for a discussion of similar considerations in the context of dimension reduction techniques.

**The Manifold Hypothesis.** The manifold hypothesis asserts that nominally high-dimensional data from the real world are concentrated on or near a low-dimensional set embedding in high-dimensional space (Cayton, 2005), Bengio et al. (2013); Fefferman et al. (2016). This phenomenon is the motivation for intrinsic dimension estimation methods, e.g., (Kégl, 2002; Levina & Bickel, 2004;?; Hein & Audibert, 2005; Carter et al., 2009; Little, 2011) and wide range of nonlinear dimension reduction techniques, including: Tenenbaum et al. (2000); Van der Maaten & Hinton (2008); McInnes et al. (2018).

Whiteley et al. (2026) proposed a generic form of latent variable model to explain manifold structure in data. The random function model considered in the present paper is a variation on the model of Whiteley et al. (2026), with the following differences:

- In the present paper we parameterise our model in terms of the covariance matrices $\mathbf{\Sigma}(\cdot)$ and mean vectors $\boldsymbol{\mu}(\cdot)$, and it is this parameterization which enables us to explain the role of the effective dimensions $p_{\text{eff}}^{(i)}$. Whiteley et al. (2026) did not consider this parameterization or the notion of effective dimension.

- Connected with this parameterization, the kernel function we consider (4) differs from that of Whiteley et al. (2026) in how it is normalised.

- Whiteley et al. (2026) studied the statistical properties of the PCA embedding of data from their model, under assumptions including: i) finite rank ($r < \infty$), ii) independence across features, iii) uniformly bounded fourth moments, and iv) latent variables $z_i$ being random, and iid. By contrast our Theorem 3.1 and Proposition 3.2 address not the PCA embedding for rather dot-products between data vectors, we make no assumption about $r$ being finite, we assume sub-Gaussianity, and do not make any statistical assumptions about the $z_i$. In this sense the assumptions of the present work are in some ways more general (we do not require independence) but in other ways stronger and leading to more refined results (we require sub-Gaussianity, and that leads to our $\sqrt{\log n / p_{\text{eff}}}$ convergence rates.)

Whiteley et al. (2026) explained homeomorphic (as per Lemma 3.3) and isometric properties of kernel feature maps, including giving sufficient conditions on the kernel function for isometry to hold, building on older work, Whiteley et al. (2021).

Motivated by hierarchical clustering Gray et al. (2023) studied asymptotic behaviour of dot-products between data vectors as a function of $n$ and $p$ under a tree structured model. They assumed mixing across dimensions and a polynomial moment condition. The convergence rates they obtain are slower than those we obtain, reflecting our sub-Gaussian assumptions. The notion of effective dimension does not appear in Gray et al. (2023).

**The Hanson-Wright inequality.** The Hanson-Wright (HW) inequality is a concentration inequality for quadratic forms $\mathbf{X}^\top \mathbf{A} \mathbf{X}$, where $\mathbf{X}$ is a vector of independent, sub-Gaussian random variables. A version of this theorem was first published in (Hanson & Wright, 1971; Wright, 1973). The result has been revisited under various sets of assumptions surveyed by Rudelson & Vershynin (2013, Rem. 1.2), who presented their own proof using tools from high-dimensional probability. Our GHW inequality concerns concentration of $\mathbf{X}^\top \mathbf{A} \mathbf{X}'$, where the random vectors $\mathbf{X} = (X_1, \ldots, X_p)$ and $\mathbf{X}' = (X_1', \ldots, X_p')$ are such that the pairs $(X_j, X_j')$ are independent across $j = 1, \ldots, p$. Our proof follows the strategy of Rudelson & Vershynin (2013, Thm 1.1) very closely and our inequality reduces to exactly their result when $\mathbf{X} = \mathbf{X}'$. At a technical level, the generality achieved in our proof is therefore modest, but in terms of interpretation and applicability, our result presents a substantial step forward because it opens the door to understanding the concentration behaviour of dot products among point clouds using the GHW inequality.

**HDLSS asymptotics.** The theory of high-dimension, low sample size (HDLSS) asymptotics Hall et al. (2005); Ahn et al. (2007); Shen et al. (2016); Aoshima et al. (2018) has made a transformative impact on our understanding and intuition for how random vectors behave in high-dimensions. The earliest of these works treated the case where $n$ is fixed and $p \to \infty$, whilst noting that the case $p/n^2 \to \infty$ could be treated in a similar fashion. This reflects the 4-th moment conditions assumed in Hall et al. (2005), which is considerably weaker than sub-Gaussianity. The significance of the regime $p/n^2 \to \infty$ can be illustrated as follows.

Let $\mathbf{X} = (X_1, \ldots, X_p)$ be a zero-mean random vector with i.i.d elements satisfying $\mathbb{E}[|X_j|^2] = 1$ and $\mathbb{E}[|X_j|^4] < \infty$, and let $\mathbf{X}'$ be an independent copy of $\mathbf{X}$. Then $\mathbb{E}[\|\mathbf{X}\|^2] = p$ and $\mathbb{E}[\mathbf{X} \cdot \mathbf{X}'] = 0$. Noting that $\|\mathbf{X}\|^2 - p$ and $\mathbf{X} \cdot \mathbf{X}'$ are both sums of independent, mean-zero random variables, Chebychev's inequality implies:

$$\max\left\{\mathbb{P}\left(\left|\frac{\|\mathbf{X}\|^2}{p} - 1\right| > \delta\right), \mathbb{P}\left(\left|\frac{\mathbf{X} \cdot \mathbf{X}'}{p}\right| > \delta\right)\right\} \leq \frac{\mathbb{E}[|X_1|^4]}{\delta^2 p}.$$

Via a union bound, it follows that if $\mathbf{X}_1, \ldots, \mathbf{X}_n$ are independent copies of $\mathbf{X}$, then

$$\max_{i,j \in [n]} \left|\frac{1}{p}\mathbf{X}_i \cdot \mathbf{X}_j - \mathbf{I}[i = j]\right| \in O_{\mathbb{P}}\left(\frac{n}{\sqrt{p}}\right)$$

for $p$ growing as $n \to \infty$, cf. Proposition 2.2. The sub-Gaussian setting of Proposition 2.2 is far stronger than the finite fourth moment condition invoked above. The price to pay for the finite fourth moment condition is the slower $n/\sqrt{p}$ convergence rate.

**Under sub-Gaussian assumptions, when does $p \gg n$ really matter?** The convergence rates we obtain in the present paper are closely tied to the combination of our sub-Gaussian assumptions and the $\max_{i,j \in [n]}$ appearing in Proposition 2.2 and Theorem 3.1. Let use define the matrix $\mathbf{Y} = [\mathbf{Y}_1 | \cdots | \mathbf{Y}_n]^\top \in \mathbb{R}^{n \times n}$. Assuming for ease of exposition that $\mathbb{E}[\|\mathbf{Y}_i\|^2] = p$ and, our results about dot products $\mathbf{Y}_i \cdot \mathbf{Y}_j$ as in Proposition 2.2 and Theorem 3.1 can be re-written as controlling $p^{-1}\|\mathbf{Y}\mathbf{Y}^\top - \mathbb{E}[\mathbf{Y}\mathbf{Y}^\top]\|_{\max}$, where for a matrix $\mathbf{A} = (A_{ij})$, $\|\mathbf{A}\|_{\max} = \max_{ij} |A_{ij}|$.

Under sub-Gaussian assumptions, other matrix norms of $\mathbf{Y}\mathbf{Y}^\top - \mathbb{E}[\mathbf{Y}\mathbf{Y}^\top]$ can scale differently with $n$ and $p$, notably the spectral norm. For example, with $\tilde{\mathbf{Y}}_j$ denoting the $j$th column of $\mathbf{Y}$ so that $p^{-1}\mathbf{Y}\mathbf{Y}^\top \equiv p^{-1}\sum_{j=1}^p \tilde{\mathbf{Y}}_j \tilde{\mathbf{Y}}_j^\top$, if the vectors $\tilde{\mathbf{Y}}_j$ are i.i.d. and uniformly bounded as $\|\tilde{\mathbf{Y}}_j\|^2 \leq \text{const.} \cdot n$, then by Tropp et al. (2015, Sec 1.6.3.), $p \sim n \log n$ is sufficient to control the relative error $\mathbb{E}[\|\mathbf{Y}\mathbf{Y}^\top - \mathbb{E}[\mathbf{Y}\mathbf{Y}^\top]\|]/\|\mathbb{E}[\mathbf{Y}\mathbf{Y}^\top]\|$, and this condition is, in the words of Tropp et al. (2015, Sec 1.6.3.), "qualitatively sharp for worst case distributions".

We note that the condition "$p \gg n$" is also famously associated with sparse regression problems in which the number of available covariates, $p$, is much larger than the number of samples, $n$, as can be approached using the Lasso (Tibshirani, 1996) and similar techniques. Those sparse regression problems seem not to be closely related to the geometric considerations in the present work.

## B. Empirical degradation outside the sub-Gaussian regime

In the main text, our concentration inequalities (e.g. Proposition 2.2) rely on the assumption that the noise variables are sub-Gaussian. As noted in Section 7, relaxing this assumption to sub-exponential or polynomial moment conditions would result in slower convergence rates.

To empirically illustrate this degradation, for a range of distributions with varying tail heaviness, we evaluate $\max_{i \neq j} \frac{|Y_i \cdot Y_j|}{p}$ as the sample size $n$ increases, with a fixed ambient dimension $p = 100$. To ensure a fair comparison, all data generating distributions are standardised to have zero mean and identity covariance. We evaluate four distributions, representing a range of tail behaviours:

1. **Gaussian**: A standard sub-Gaussian baseline.

2. **Laplace**: A sub-exponential distribution.

3. **Student's $t$ (df=5)**: A heavy-tailed distribution with a finite fourth moment.

4. **Student's $t$ (df=3)**: A heavy-tailed distribution with an infinite fourth moment.

We vary $n$ logarithmically from $n = 10$ to $n \approx 3000$ and record the maximum inner product across all pairs. The reported values for each $n$ are averaged over 10 independent trials, and are shown in Figure 5. As expected, the sub-Gaussian distribution exhibits slow growth, whereas the heavier tails grow faster. For the Student's $t$-distribution with df=3, the maximum inner product diverges rapidly.

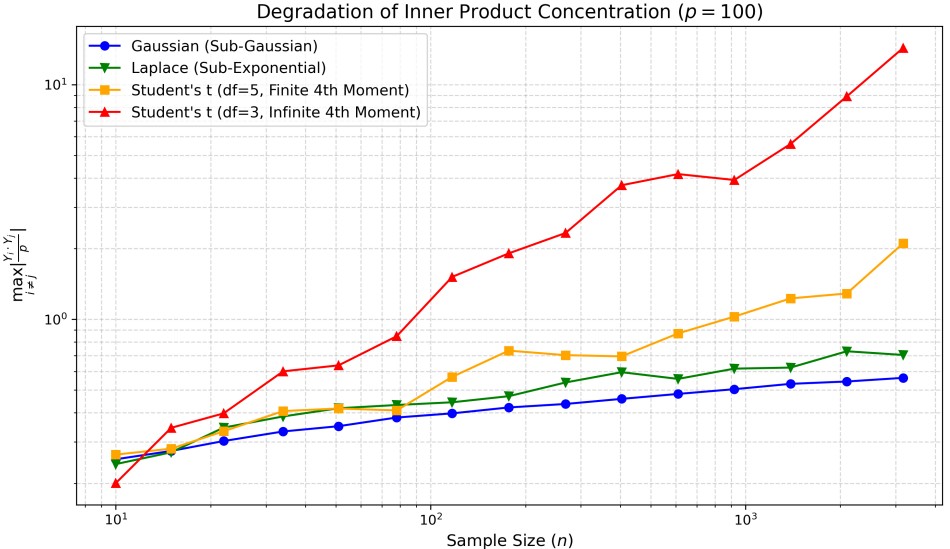

*Figure 5.* $\max_{i \neq j} \frac{|Y_i \cdot Y_j|}{p}$ for $n$ independent samples in $\mathbb{R}^{100}$. All distributions are scaled to have mean-zero and identity-covariance. Sub-Gaussian (blue) and sub-exponential (green) distributions grow very slowly with $n$. Relaxing to a finite fourth moment (orange, Student's $t$ df=5) results in slightly faster growth, and distributions with an infinite fourth moment (red, Student's $t$ df=3) diverge rapidly.

## C. Proofs and supporting results for Section 2

The sub-exponential norm of a random variable $X$ is:

$$\|X\|_{\psi_1} := \sup_{q \geq 1} q^{-1} \mathbb{E}[|X|^q]^{1/q}.$$

*Proof of Theorem 2.1.* The proof follows the arguments of Rudelson & Vershynin (2013, Proof of thm 1.1) very closely. As they did, by replacing $\mathbf{X}$ with $\mathbf{X}/K$ we can assume without loss of generality that $K = 1$.

Our first objective is to estimate

$$\pi := \mathbb{P}\left(\mathbf{X}^\top \mathbf{A} \mathbf{X}' - \mathbb{E}\left[\mathbf{X}^\top \mathbf{A} \mathbf{X}'\right] > t\right).$$

Let $\mathbf{A} = (a_{ij})_{i,j=1}^p$. The independence assumption of the theorem implies that $X_i$ and $X'_j$ are independent for $i \neq j$. Using this together with the zero-mean assumption of the theorem,

$$\mathbf{X}^\top \mathbf{A} \mathbf{X}' - \mathbb{E}\left[\mathbf{X}^\top \mathbf{A} \mathbf{X}'\right] = \sum_{i,j} a_{ij} X_i X'_j - \sum_i a_{ii} \mathbb{E}[X_i X'_i]$$

$$= \sum_i a_{ii}\left(X_i X'_i - \mathbb{E}[X_i X'_i]\right) + \sum_{i \neq j} a_{ij} X_i X'_j.$$

The problem is split up into estimating deviation probabilities associated with the diagonal and off-diagonal sums:

$$\pi \leq \mathbb{P}\left(\sum_i a_{ii}\left(X_i X'_i - \mathbb{E}[X_i X'_i]\right) > t/2\right) + \mathbb{P}\left(\sum_{i \neq j} a_{ij} X_i X'_j > t/2\right) =: \pi_1 + \pi_2.$$

Note that $X_i X_i' - \mathbb{E}[X_i X_i']$ are independent, mean-zero, sub-exponential random variables, indeed for $q \geq 1$, by Minkowski's, Jensen's and the Cauchy-Schwartz inequalities,

$$\mathbb{E}\left[|X_i X_i' - \mathbb{E}[X_i X_i']|^q\right]^{1/q} \leq \mathbb{E}\left[|X_i X_i'|^q\right]^{1/q} + |\mathbb{E}[X_i X_i']|$$

$$\leq 2\mathbb{E}\left[|X_i X_i'|^q\right]^{1/q} \leq 2\mathbb{E}\left[|X_i|^{2q}\right]^{1/2q} \mathbb{E}\left[|X_i'|^{2q}\right]^{1/2q}$$

and hence

$$\|X_i X_i' - \mathbb{E}[X_i X_i']\|_{\psi_1} \leq 2\|X_i X_i'\|_{\psi_1} \leq 4\|X_i\|_{\psi_2}\|X_i'\|_{\psi_2} \leq 4.$$

By a Bernstein-type inequality Vershynin (2010, Prop 5.16),

$$\pi_1 \leq \exp\left[-c\min\left\{\frac{t^2}{\|\mathbf{A}\|_{\mathrm{F}}^2}, \frac{t}{\|\mathbf{A}\|}\right\}\right].$$

It remains to consider the term:

$$S := \sum_{i \neq j} a_{ij} X_i X_j'.$$

For $\lambda > 0$ we have

$$\pi_2 = \mathbb{P}\left(S > t/2\right) \leq \exp(-\lambda t/2)\mathbb{E}\left[\exp(\lambda S)\right].$$

Let $\delta_1, \ldots, \delta_p$ be independent $\{0,1\}$-valued random variables with $\mathbb{E}[\delta_i] = 1/2$. Then

$$S = 4\mathbb{E}_\delta[S_\delta], \quad \text{where} \quad S_\delta := \sum_{i,j} \delta_i(1 - \delta_j)a_{ij} X_i X_j',$$

and where $\mathbb{E}_\delta$ denotes expectation with respect the distribution of $\delta = (\delta_1, \ldots, \delta_p)$. By Jensen's inequality,

$$\mathbb{E}\left[\exp(\lambda S)\right] \leq \mathbb{E}\left[\exp(4\lambda S_\delta)\right].$$

With $\Lambda_\delta := \{i \in [p] : \delta_i = 1\}$, we have

$$S_\delta = \sum_{j \in \Lambda_\delta^c} X_j'\left(\sum_{i \in \Lambda_\delta} a_{ij} X_i\right).$$

Conditional on $\delta$ and $(X_i)_{i \in \Lambda_\delta}$, $S_\delta$ is a linear combination of mean-zero, sub-Gaussian random variables $X_j'$, $j \in \Lambda_\delta^c$, with associated coefficients $\sum_{i \in \Lambda_\delta} a_{ij} X_i$. It follows that the conditional distribution of $S_\delta$ is sub-Gaussian with (conditional) sub-Gaussian norm:

$$\|S_\delta\|_{\psi_1} \leq C\sigma_\delta, \qquad \sigma_\delta^2 := \sum_{j \in \Lambda_\delta^c}\left(\sum_{i \in \Lambda_\delta} a_{ij} X_i\right)^2,$$

for some constant $C > 0$.

Noting that $\sigma_\delta^2$ is a function of $\mathbf{X}$ but not of $\mathbf{X}'$, the remaining steps in the proof follow the same arguments as in Rudelson & Vershynin (2013, Proof of thm 1.1), so the details are omitted. $\qquad\square$

*Proof of Proposition 2.2.* Under the assumptions of the proposition we have $\mathbb{E}[\|\mathbf{Y}_i\|^2] = \mathrm{tr}\boldsymbol{\Sigma}$ and $\mathbb{E}[\mathbf{Y}_i \cdot \mathbf{Y}_j] = 0$ for $i \neq j$, hence

$$\pi_{ij}(t) := \mathbb{P}\left(\left|\frac{\mathbf{Y}_i \cdot \mathbf{Y}_j}{\mathrm{tr}\boldsymbol{\Sigma}} - \mathbf{I}[i = j]\right| > t\right) = \mathbb{P}\left(|\mathbf{X}_i^\top \boldsymbol{\Sigma}\mathbf{X}_j - \mathbb{E}[\mathbf{X}_i^\top \boldsymbol{\Sigma}\mathbf{X}_j]| > t \cdot \mathrm{tr}\boldsymbol{\Sigma}\right).$$

Applying Theorem 2.1 gives

$$\pi_{ij}(t) \leq 2\exp\left[-c\min\left\{\frac{t^2(\mathrm{tr}\boldsymbol{\Sigma})^2}{K^4\mathrm{tr}(\boldsymbol{\Sigma}^2)}, \frac{t \cdot \mathrm{tr}\boldsymbol{\Sigma}}{K^2\|\boldsymbol{\Sigma}\|}\right\}\right],$$

and by a union bound

$$
\mathbb{P}\left(\max_{i,j\in[n]}\left|\frac{\mathbf{Y}_i\cdot\mathbf{Y}_j}{\mathrm{tr}\mathbf{\Sigma}}-\mathbf{I}[i=j]\right|>t\right)
$$

$$
\leq\sum_{i,j\in[n]}\pi_{ij}(t)
$$

$$
\leq 2n^2\exp\left[-c\min\left\{\frac{t^2(\mathrm{tr}\mathbf{\Sigma})^2}{K^4\mathrm{tr}(\mathbf{\Sigma}^2)},\frac{t\cdot\mathrm{tr}\mathbf{\Sigma}}{K^2\|\mathbf{\Sigma}\|}\right\}\right]
$$

$$
= 2\exp\left[-c\min\left\{\frac{t^2(\mathrm{tr}\mathbf{\Sigma})^2}{K^4\mathrm{tr}(\mathbf{\Sigma}^2)},\frac{t\cdot\mathrm{tr}\mathbf{\Sigma}}{K^2\|\mathbf{\Sigma}\|}\right\}+2\log n\right]. \tag{9}
$$

Let $M>0$ be a constant whose value will be chosen later. Applying (9) with $t=\sqrt{\frac{\log n}{p_{\mathrm{eff}}}}M$,

$$
\mathbb{P}\left(\max_{i,j\in[n]}\left|\frac{\mathbf{Y}_i\cdot\mathbf{Y}_j}{\mathrm{tr}\mathbf{\Sigma}}-\mathbf{I}[i=j]\right|>\sqrt{\frac{\log n}{p_{\mathrm{eff}}}}M\right)
$$

$$
\leq 2\exp\left[-c(\log n)\frac{M}{K^2}\min\left\{\frac{M(\mathrm{tr}\mathbf{\Sigma})^2}{K\mathrm{tr}(\mathbf{\Sigma}^2)}\frac{\|\mathbf{\Sigma}\|}{\mathrm{tr}\mathbf{\Sigma}},\sqrt{\frac{\mathrm{tr}\mathbf{\Sigma}}{\|\mathbf{\Sigma}\|}}\frac{1}{\sqrt{\log n}}\right\}+2\log n\right]. \tag{10}
$$

We have:

$$
\frac{(\mathrm{tr}\mathbf{\Sigma})^2}{\mathrm{tr}(\mathbf{\Sigma}^2)}\frac{\|\mathbf{\Sigma}\|}{\mathrm{tr}\mathbf{\Sigma}}=\frac{(\mathrm{tr}\mathbf{\Sigma})}{\mathrm{tr}(\mathbf{\Sigma}^2)}\|\mathbf{\Sigma}\|=\frac{\lambda_1\sum_i\lambda_i}{\sum_i\lambda_i^2}\geq 1,
$$

where $\lambda_1\geq\lambda_2\geq\cdots$ are the eigenvalues of $\mathbf{\Sigma}$. Using the assumption that $\mathrm{tr}(\mathbf{\Sigma})/\|\mathbf{\Sigma}\|\in\Omega(\log n)$, there exists $c_0>0$ and $n_0\geq 1$ such that for $n\geq n_0$,

$$
\sqrt{\frac{\mathrm{tr}\mathbf{\Sigma}}{\|\mathbf{\Sigma}\|}}\frac{1}{\sqrt{\log n}}\geq c_0.
$$

Hence for $n\geq n_0$,

$$
\min\left\{\frac{M(\mathrm{tr}\mathbf{\Sigma})^2}{K\mathrm{tr}(\mathbf{\Sigma}^2)}\frac{\|\mathbf{\Sigma}\|}{\mathrm{tr}\mathbf{\Sigma}},\sqrt{\frac{\mathrm{tr}\mathbf{\Sigma}}{\|\mathbf{\Sigma}\|}}\frac{1}{\sqrt{\log n}}\right\}\geq\min\left\{\frac{M}{K},c_0\right\}. \tag{11}
$$

Now fix any $\epsilon>0$. By choosing $M$ large enough that $M/K\geq c_0$, using (11), and then, increasing $M$ if necessary. we can achieve:

$$
2\exp\left[-c(\log n_0)\frac{M}{K^2}\min\left\{\frac{M}{K},c_0\right\}+2\log n_0\right]
$$

$$
\leq 2\exp\left[-\log n_0\left(\frac{M}{K^2}cc_0-2\right)\right]
$$

$$
\leq\epsilon.
$$

Combining this inequality with (10) and (11), we have for any $n\geq n_0$,

$$
\mathbb{P}\left(\max_{i,j\in[n]}\left|\frac{\mathbf{Y}_i\cdot\mathbf{Y}_j}{\mathrm{tr}\mathbf{\Sigma}}-\mathbf{I}[i=j]\right|>\sqrt{\frac{\log n}{p_{\mathrm{eff}}}}M\right)\leq\epsilon,
$$

which completes the proof. $\qquad\square$

## D. Proofs and supporting results for Section 3

Throughout the proofs in this section we will repeatedly use the fact that the square root of a symmetric, positive semidefinite matrix $\mathbf{A}$ is a symmetric, positive semidefinite matrix $\mathbf{B}$ such that $\mathbf{BB}=\mathbf{BB}^\top=\mathbf{A}$.

From the definition of $\mathbf{Y}_i$ in Section 3 and **A2**, we have that

$$\mathbb{E}[\|\mathbf{Y}_i\|^2] = \text{tr}(\mathbf{\Sigma}(z_i)) + \|\boldsymbol{\mu}(z_i)\|^2 + p\sigma^2,$$

and

$$\mathbb{E}[\mathbf{Y}_i \cdot \mathbf{Y}_j] = \mathbb{E}[\mathbf{Y}^{\text{nf}}(z_i) \cdot \mathbf{Y}^{\text{nf}}(z_j)] + \mathbf{I}[i = j]p\sigma^2.$$

Let us define

$$
\begin{aligned}
C_{ij} &:= \mathbb{E}[\|\mathbf{Y}_i\|^2]^{1/2}\mathbb{E}[\|\mathbf{Y}_j\|^2]^{1/2} \\
&= \left(\text{tr}(\mathbf{\Sigma}(z_i)) + \|\boldsymbol{\mu}(z_i)\|^2 + p\sigma^2\right)^{1/2}\left(\text{tr}(\mathbf{\Sigma}(z_j)) + \|\boldsymbol{\mu}(z_j)\|^2 + p\sigma^2\right)^{1/2},
\end{aligned}
$$

and $\mathbf{\Sigma}_{ij} := \mathbf{\Sigma}(z_i)^{1/2}\mathbf{\Sigma}(z_j)^{1/2}$ (N.B., for each $i, j$, $\mathbf{\Sigma}_{ij}$ is a matrix).

We use the following lemmas D.1-D.5 in our proof of Theorem 3.1. These lemmas make rely on the same assumptions as Theorem 3.1, but to avoid repetition we do not explicitly refer to these assumptions in the statements of the lemmas.

In the statements and proofs of lemmas D.1-D.5, $M > 0$ is an arbitrarily chosen constant and

$$t := \sqrt{\frac{\log n}{\min_{i \in [n]}\left(p_{\text{eff}}^{(i)}\right)}}M.$$

**Lemma D.1.** *There exists $c_1, c_2 > 0$ and $n_0$ such that for $n \geq n_0$,*

$$\mathbb{P}\left(\left|\mathbf{X}(z_i)^\top\mathbf{\Sigma}_{ij}\mathbf{X}(z_j) - \mathbb{E}[\mathbf{X}(z_i)^\top\mathbf{\Sigma}_{ij}\mathbf{X}(z_j)]\right| > \frac{t}{8}C_{ij}\right)$$

$$\leq 2\exp\left(-(\log n)\frac{c_1 M}{K^2}\min\left\{\frac{M}{K^2}, c_2\right\}\right)$$

*for all $i, j \in [n]$.*

*Proof.* We can bound the probability above using Theorem 2.1 with $\mathbf{A} = \mathbf{\Sigma}_{ij}$, noting $\mathbf{\Sigma}(z_i)^{1/2}, \mathbf{\Sigma}(z_j)^{1/2}$ are symmetric matrices, and using the cyclicity of trace, $\|\mathbf{\Sigma}_{ij}\|_{\text{F}}^2 = \text{tr}(\mathbf{\Sigma}_{ij}^\top\mathbf{\Sigma}_{ij}) = \text{tr}[\mathbf{\Sigma}(z_i)\mathbf{\Sigma}(z_j)]$, to get

$$\mathbb{P}\left(\left|\mathbf{X}(z_i)^\top\mathbf{\Sigma}_{ij}\mathbf{X}(z_j) - \mathbb{E}[\mathbf{X}(z_i)^\top\mathbf{\Sigma}_{ij}\mathbf{X}(z_j)]\right| > \frac{t}{8}C_{ij}\right)$$

$$\leq 2\exp\left(-c_1\min\left\{\frac{t^2 C_{ij}^2}{K^4\text{tr}[\mathbf{\Sigma}(z_i)\mathbf{\Sigma}(z_j)]}, \frac{tC_{ij}}{K^2\|\mathbf{\Sigma}_{ij}\|}\right\}\right). \tag{12}$$

Then, focusing on the first term inside the minimum in (12), we use $C_{ij} \geq \text{tr}(\mathbf{\Sigma}(z_i))^{1/2}\text{tr}(\mathbf{\Sigma}(z_j))^{1/2}$, $\text{tr}[\mathbf{\Sigma}(z_i)\mathbf{\Sigma}(z_j)] \leq \left[\text{tr}(\mathbf{\Sigma}(z_i)^2)\text{tr}(\mathbf{\Sigma}(z_j)^2)\right]^{1/2}$ and the following bound on $t$,

$$t \geq \sqrt{\frac{\log n}{\left(p_{\text{eff}}^{(i)}p_{\text{eff}}^{(j)}\right)^{1/2}}}M = \sqrt{\frac{\log n\|\mathbf{\Sigma}(z_i)\|^{1/2}\|\mathbf{\Sigma}(z_j)\|^{1/2}}{\text{tr}(\mathbf{\Sigma}(z_i))^{1/2}\text{tr}(\mathbf{\Sigma}(z_j))^{1/2}}}M, \tag{13}$$

to obtain:

$$
\begin{aligned}
\frac{t^2 C_{ij}^2}{K^4\text{tr}[\mathbf{\Sigma}(z_i)\mathbf{\Sigma}(z_j)]} &\geq \frac{(\log n)M^2}{K^4}\frac{\|\mathbf{\Sigma}(z_i)\|^{1/2}\|\mathbf{\Sigma}(z_j)\|^{1/2}\text{tr}(\mathbf{\Sigma}(z_i))^{1/2}\text{tr}(\mathbf{\Sigma}(z_j))^{1/2}}{\text{tr}[\mathbf{\Sigma}(z_i)\mathbf{\Sigma}(z_j)]} \\
&\geq \frac{(\log n)M^2}{K^4}\sqrt{\frac{\lambda_1^{(i)}\lambda_1^{(j)}\left(\sum_k \lambda_k^{(i)}\right)\left(\sum_k \lambda_k^{(j)}\right)}{\left(\sum_k \lambda_k^{{(i)}^2}\right)\left(\sum_k \lambda_k^{{(j)}^2}\right)}} \\
&\geq \frac{(\log n)M^2}{K^4}, \tag{14}
\end{aligned}
$$

where $(\lambda_k^{(i)})_{k\geq 1}$ and $(\lambda_k^{(j)})_{k\geq 1}$ are the eigenvalues of $\boldsymbol{\Sigma}(z_i)$ and $\boldsymbol{\Sigma}(z_j)$ respectively.

Now, for the second term inside the minimum in (12), using that

$$C_{ij} \geq \mathrm{tr}(\boldsymbol{\Sigma}(z_i))^{1/2}\mathrm{tr}(\boldsymbol{\Sigma}(z_j))^{1/2} \geq \mathrm{tr}(\boldsymbol{\Sigma}_{ij})$$

, $\|\boldsymbol{\Sigma}(z_i)\|^{1/2}\|\boldsymbol{\Sigma}(z_j)\|^{1/2} \geq \|\boldsymbol{\Sigma}_{ij}\|$ and the bound on $t$ in (13), we get

$$\begin{aligned}
\frac{tC_{ij}}{K^2\|\boldsymbol{\Sigma}_{ij}\|} &\geq \frac{\mathrm{tr}(\boldsymbol{\Sigma}(z_i))^{1/2}\mathrm{tr}(\boldsymbol{\Sigma}(z_j))^{1/2}M}{K^2\|\boldsymbol{\Sigma}_{ij}\|}\sqrt{\frac{\log n\|\boldsymbol{\Sigma}(z_i)\|^{1/2}\|\boldsymbol{\Sigma}(z_j)\|^{1/2}}{\mathrm{tr}(\boldsymbol{\Sigma}(z_i))^{1/2}\mathrm{tr}(\boldsymbol{\Sigma}(z_i))^{1/2}}} \\
&\geq \frac{\sqrt{\log n}M}{K^2}\sqrt{\frac{\mathrm{tr}(\boldsymbol{\Sigma}(z_i))^{1/2}\mathrm{tr}(\boldsymbol{\Sigma}(z_j))^{1/2}}{\|\boldsymbol{\Sigma}(z_i)\|^{1/2}\|\boldsymbol{\Sigma}(z_j)\|^{1/2}}}.
\end{aligned} \tag{15}$$

Putting together (14) and (15), we have

$$\begin{aligned}
\min&\left\{\frac{t^2C_{ij}^2}{K^4\mathrm{tr}[\boldsymbol{\Sigma}(z_i)\boldsymbol{\Sigma}(z_j)]}, \frac{tC_{ij}}{K^2\|\boldsymbol{\Sigma}_{ij}\|}\right\} \\
&\geq (\log n)\frac{M}{K^2}\min\left\{\frac{M}{K^2}, \sqrt{\frac{\mathrm{tr}(\boldsymbol{\Sigma}(z_i))^{1/2}\mathrm{tr}(\boldsymbol{\Sigma}(z_j))^{1/2}}{\|\boldsymbol{\Sigma}(z_i)\|^{1/2}\|\boldsymbol{\Sigma}(z_j)\|^{1/2}}}\frac{1}{\sqrt{\log n}}\right\}.
\end{aligned}$$

Using the assumption that $\min_{i\in[n]}\mathrm{tr}(\boldsymbol{\Sigma}(z_i))/\|\boldsymbol{\Sigma}(z_i)\| \in \Omega(\log n)$, there exists $c_2 > 0$ and $n_0$ such that for $n \geq n_0$,

$$\sqrt{\frac{\mathrm{tr}(\boldsymbol{\Sigma}(z_i))^{1/2}\mathrm{tr}(\boldsymbol{\Sigma}(z_j))^{1/2}}{\|\boldsymbol{\Sigma}(z_i)\|^{1/2}\|\boldsymbol{\Sigma}(z_j)\|^{1/2}}}\frac{1}{\sqrt{\log n}} \geq c_2.$$

Hence, for $n \geq n_0$,

$$\min\left\{\frac{t^2C_{ij}^2}{K^4\mathrm{tr}(\boldsymbol{\Sigma}_{ij}^2)}, \frac{tC_{ij}}{K^2\|\boldsymbol{\Sigma}_{ij}\|}\right\} \geq (\log n)\frac{M}{K^2}\min\left\{\frac{M}{K^2}, c_2\right\},$$

and substituting this into (12) the result of the lemma follows. $\qquad\square$

**Lemma D.2.** *There exists $c_3 > 0$ and $n_1$ such that for $n \geq n_1$*

$$\mathbb{P}\left(\left|\boldsymbol{\mu}(z_i)^\top\boldsymbol{\Sigma}(z_j)^{1/2}\mathbf{X}(z_j)\right| > \frac{t}{8}C_{ij}\right) \leq 2\exp\left(-\frac{c_3M^2}{K^2}(\log n)\right).$$

*for all $i, j \in [n]$.*

*Proof.* Since $\boldsymbol{\mu}(z_i)^\top\boldsymbol{\Sigma}(z_j)^{1/2}\mathbf{X}(z_j)$ is the sum of independent centered sub-gaussian random variables, its squared sub-gaussian norm is bounded

$$\|\boldsymbol{\mu}(z_i)^\top\boldsymbol{\Sigma}(z_j)^{1/2}\mathbf{X}(z_j)\|_{\psi_2}^2 \leq K^2\|\boldsymbol{\mu}(z_i)^\top\boldsymbol{\Sigma}(z_j)^{1/2}\|_2^2 = K^2\boldsymbol{\mu}(z_i)^\top\boldsymbol{\Sigma}(z_j)\boldsymbol{\mu}(z_i).$$

Therefore we can use a Hoeffding-type inequality for sub-gaussian random variables (see e.g. Vershynin (2010, Prop. 5.10) or Vershynin (2018, Thm 2.6.3)) to get

$$\mathbb{P}\left(\left|\boldsymbol{\mu}(z_i)^\top\boldsymbol{\Sigma}(z_j)^{1/2}\mathbf{X}(z_j)\right| > \frac{t}{8}C_{ij}\right) \leq 2\exp\left(-\frac{c_4}{K^2}\frac{t^2C_{ij}^2}{\boldsymbol{\mu}(z_i)^\top\boldsymbol{\Sigma}(z_j)\boldsymbol{\mu}(z_i)}\right). \tag{16}$$

Now, we use following lower bound on $C_{ij}^2$,

$$C_{ij}^2 \geq \mathrm{tr}(\boldsymbol{\Sigma}(z_j))\|\boldsymbol{\mu}(z_i)\|_2^2,$$

together with

$$t \geq \sqrt{\frac{\log n}{\left(p_{\text{eff}}^{(i)} p_{\text{eff}}^{(j)}\right)^{1/2}}} M = \sqrt{\frac{\log n \|\mathbf{\Sigma}(z_i)\|^{1/2}\|\mathbf{\Sigma}(z_j)\|^{1/2}}{\text{tr}(\mathbf{\Sigma}(z_i))^{1/2}\text{tr}(\mathbf{\Sigma}(z_j))^{1/2}}} M,$$

to get

$$\frac{t^2 C_{ij}^2}{\boldsymbol{\mu}(z_i)^\top \mathbf{\Sigma}(z_j)\boldsymbol{\mu}(z_i)} \geq (\log n) M^2 \frac{\|\mathbf{\Sigma}(z_i)\|^{1/2}\|\mathbf{\Sigma}(z_j)\|^{1/2}\text{tr}(\mathbf{\Sigma}(z_j))\|\boldsymbol{\mu}(z_i)\|_2^2}{[\text{tr}(\mathbf{\Sigma}(z_i))\text{tr}(\mathbf{\Sigma}(z_j))]^{1/2}\boldsymbol{\mu}(z_i)^\top \mathbf{\Sigma}(z_j)\boldsymbol{\mu}(z_i)}$$

$$= (\log n) M^2 \left(\frac{\|\mathbf{\Sigma}(z_i)\|}{\text{tr}(\mathbf{\Sigma}(z_i))}\right)^{1/2}\left(\frac{\text{tr}(\mathbf{\Sigma}(z_j))}{\|\mathbf{\Sigma}(z_j)\|}\right)^{1/2}\frac{\|\mathbf{\Sigma}(z_j)\|\|\boldsymbol{\mu}(z_i)\|_2^2}{\sum_k \left\langle \boldsymbol{\mu}(z_j), U_k^{(i)}\right\rangle^2 \lambda_k^{(j)}}$$

$$\geq (\log n) M^2 \left(\frac{\|\mathbf{\Sigma}(z_i)\|}{\text{tr}(\mathbf{\Sigma}(z_i))}\right)^{1/2}\left(\frac{\text{tr}(\mathbf{\Sigma}(z_j))}{\|\mathbf{\Sigma}(z_j)\|}\right)^{1/2},$$

where $U_k^{(i)}$ is the eigenvector of $\mathbf{\Sigma}(z_i)$ associated with eigenvalue $\lambda_k^{(i)}$. Therefore, using the assumption that $\min_{i\in[n]}\text{tr}(\mathbf{\Sigma}(z_i))/\|\mathbf{\Sigma}(z_i)\| \in \Omega(\log n)$, there exists $c_3 > 0$ and $n_1$ such that for $n \geq n_1$,

$$\frac{t^2 C_{ij}^2}{\boldsymbol{\mu}(z_i)^\top \mathbf{\Sigma}(z_j)\boldsymbol{\mu}(z_i)} \geq c_3 M^2(\log n).$$

Substituting this into (16), gives the result. $\qquad\square$

**Lemma D.3.** *There exists $c_4, c_5 > 0$ and $n_2$ such that for $n \geq n_2$*

$$\mathbb{P}\left(\left|\sigma \mathbf{E}_i^\top \mathbf{\Sigma}(z_j)^{1/2}\mathbf{X}(z_j)\right| > \frac{t}{8}C_{ij}\right) \leq 2\exp\left(-(\log n)\frac{c_5 M}{K^2}\min\left\{\frac{M}{K^2}, c_4\right\}\right)$$

*for all $i, j \in [n]$.*

*Proof.* First, we bound the above probability using Theorem 2.1 with $\mathbf{A} = \sigma \mathbf{\Sigma}(z_j)^{1/2}$ to get

$$\mathbb{P}\left(\left|\sigma \mathbf{E}_i^\top \mathbf{\Sigma}(z_j)^{1/2}\mathbf{X}(z_j)\right| > \frac{t}{8}C_{ij}\right)$$

$$\leq 2\exp\left(-c_5\min\left\{\frac{t^2 C_{ij}^2}{\sigma^2 K^4\text{tr}[\mathbf{\Sigma}(z_j)]}, \frac{tC_{ij}}{\sigma K^2\|\mathbf{\Sigma}(z_j)^{1/2}\|}\right\}\right). \quad (17)$$

Now, focusing on the first term inside the minimum, we use that $C_{ij}^2 \geq p\sigma^2\text{tr}[\mathbf{\Sigma}(z_j)]$ and $t \geq M\sqrt{(\log n)/p}$ to get

$$\frac{t^2 C_{ij}^2}{\sigma^2 K^4\text{tr}[\mathbf{\Sigma}(z_j)]} \geq \frac{M^2\log n}{K^4}.$$

Focusing now on the second term inside the minimum in (17), and using that $C_{ij} \geq p^{1/2}\sigma\text{tr}[\mathbf{\Sigma}(z_j)]^{1/2}$ and $t \geq M\sqrt{(\log n)/p}$ we get

$$\frac{tC_{ij}}{\sigma K^2\|\mathbf{\Sigma}(z_j)^{1/2}\|} \geq \frac{M(\sqrt{\log n})\text{tr}[(\mathbf{\Sigma}(z_j)]^{1/2}}{K^2\|\mathbf{\Sigma}(z_j)^{1/2}\|}$$

Therefore, we have that

$$\min\left\{\frac{t^2 C_{ij}^2}{\sigma^2 K^4\text{tr}[\mathbf{\Sigma}(z_j)]}, \frac{tC_{ij}}{\sigma K^2\|\mathbf{\Sigma}(z_j)^{1/2}\|}\right\} \geq \frac{M}{K^2}(\log n)\min\left\{\frac{M}{K^2}, \frac{\text{tr}[(\mathbf{\Sigma}(z_j)]^{1/2}}{\|\mathbf{\Sigma}(z_j)^{1/2}\|}\frac{1}{\sqrt{\log n}}\right\},$$

and using the assumption that $\min_{i\in[n]}\text{tr}(\mathbf{\Sigma}(z_i))/\|\mathbf{\Sigma}(z_i)\| \in \Omega(\log n)$, there exists $c_4 > 0$ and $n_2$ such that for $n \geq n_2$,

$$\frac{\text{tr}[(\mathbf{\Sigma}(z_j)]^{1/2}}{\|\mathbf{\Sigma}(z_j)^{1/2}\|}\frac{1}{\sqrt{\log n}} \geq c_4.$$

Hence, for $n \geq n_2$,

$$\min\left\{\frac{t^2 C_{ij}^2}{\sigma^2 K^4 \mathrm{tr}[\boldsymbol{\Sigma}(z_j)]}, \frac{t C_{ij}}{\sigma K^2 \|\boldsymbol{\Sigma}(z_j)^{1/2}\|}\right\} \geq (\log n)\frac{M}{K^2}\min\left\{\frac{M}{K^2}, c_4\right\}.$$

Substituting this into (17) gives the result. $\square$

**Lemma D.4.** *For all $i, j \in [n]$ :*

$$\mathbb{P}\left(\left|\sigma \mathbf{E}_i^\top \boldsymbol{\mu}(z_j)\right| > \frac{t}{8}C_{ij}\right) \leq 2\exp\left(-\frac{c_6 M^2}{K^2}\log n\right).$$

*Proof.* Using that $\|\sigma \mathbf{E}_i^\top \boldsymbol{\mu}(z_j)\|_{\psi_2}^2 \leq \sigma^2 K^2 \|\boldsymbol{\mu}(z_j)\|_2^2$, we can bound the above probability using a Hoeffding-type equality for sub-gaussian random variables to get

$$\mathbb{P}\left(\left|\sigma \mathbf{E}_i^\top \boldsymbol{\mu}(z_j)\right| > \frac{t}{8}C_{ij}\right) \leq 2\exp\left(-\frac{c_6 t^2 C_{ij}^2}{\sigma^2 K^2 \|\boldsymbol{\mu}(z_j)\|_2^2}\right).$$

Then, using that $C_{ij}^2 \geq p\sigma^2 \|\boldsymbol{\mu}(z_j)\|_2^2$ and $t^2 \geq M^2(\log n)/p$, we get

$$\mathbb{P}\left(\left|\sigma \mathbf{E}_i^\top \boldsymbol{\mu}(z_j)\right| > \frac{t}{8}C_{ij}\right) \leq 2\exp\left(-\frac{c_6 M^2}{K^2}\log n\right).$$

$\square$

**Lemma D.5.** *There exists $c_7, c_8 > 0$ and $n_0$ such that for $n \geq n_3$*

$$\mathbb{P}\left(\left|\sigma^2(\mathbf{E}_i \cdot \mathbf{E}_j - \mathbb{E}[\mathbf{E}_i \cdot \mathbf{E}_j])\right| > \frac{t}{8}C_{ij}\right) \leq 2\exp\left(-(\log n)\frac{c_7 M}{K^2}\min\left\{\frac{M}{K^2}, c_8\right\}\right)$$

*for all $i, j \in [n]$.*

*Proof.* First, we bound the above probability using Theorem 2.1 with $\mathbf{A} = \sigma^2 \mathbf{I}_p$ to get

$$\mathbb{P}\left(\left|\sigma^2(\mathbf{E}_i \cdot \mathbf{E}_j - \mathbb{E}[\mathbf{E}_i \cdot \mathbf{E}_j])\right| > \frac{t}{8}C_{ij}\right) \leq 2\exp\left(-c_7 \min\left\{\frac{t^2 C_{ij}^2}{\sigma^4 K^4 p}, \frac{t C_{ij}}{\sigma^2 K^2}\right\}\right).$$

Then we use that $C_{ij} \geq \sigma^2 p$ and $t \geq M\sqrt{(\log n)/p}$ to get

$$\min\left\{\frac{t^2 C_{ij}^2}{\sigma^4 K^4 p}, \frac{t C_{ij}}{\sigma^2 K^2}\right\} \geq \frac{M}{K^2}(\log n)\min\left\{\frac{M}{K^2}, \frac{p^{1/2}}{\sqrt{\log n}}\right\}.$$

Finally, using that $p \geq \min_{i \in [n]} p_{\mathrm{eff}}^{(i)} \in \Omega(\log n)$, it follows that there exists $c_8 \geq 0$ and $n_3$ such that for $n \geq n_3$, $\sqrt{p/\log n} \geq c_8$. Therefore, we get that for $n \geq n_3$,

$$\mathbb{P}\left(\left|\sigma^2(\mathbf{E}_i \cdot \mathbf{E}_j - \mathbb{E}[\mathbf{E}_i \cdot \mathbf{E}_j])\right| > \frac{t}{8}C_{ij}\right) \leq 2\exp\left(-\frac{c_7 M}{K^2}(\log n)\min\left\{\frac{M}{K^2}, c_8\right\}\right).$$

$\square$

*Proof of Theorem 3.1.* Define

$$\pi_{ij}(t) := \mathbb{P}\left(\left|\frac{\mathbf{Y}_i \cdot \mathbf{Y}_j}{\mathbb{E}[\|\mathbf{Y}_i\|^2]^{1/2}\mathbb{E}[\|\mathbf{Y}_j\|^2]^{1/2}} - \frac{\mathbb{E}[\mathbf{Y}_i \cdot \mathbf{Y}_j]}{\mathbb{E}[\|\mathbf{Y}_i\|^2]^{1/2}\mathbb{E}[\|\mathbf{Y}_j\|^2]^{1/2}}\right| > t\right).$$

Using our definition of $C_{ij}$, we can rewrite this as

$$\pi_{ij}(t) = \mathbb{P}\left(|\mathbf{Y}_i \cdot \mathbf{Y}_j - \mathbb{E}[\mathbf{Y}_i \cdot \mathbf{Y}_j]| > tC_{ij}\right).$$

We can then decompose $\mathbf{Y}_i \cdot \mathbf{Y}_j - \mathbb{E}[\mathbf{Y}_i \cdot \mathbf{Y}_j]$ in the following way:

$$
\begin{aligned}
\mathbf{Y}_i &\cdot \mathbf{Y}_j - \mathbb{E}[\mathbf{Y}_i \cdot \mathbf{Y}_j] \\
&= \mathbf{X}(z_i)^\top \mathbf{\Sigma}_{ij} \mathbf{X}(z_j) - \mathbb{E}[\mathbf{X}(z_i)^\top \mathbf{\Sigma}_{ij} \mathbf{X}(z_j)] \\
&+ \boldsymbol{\mu}(z_i)^\top \mathbf{\Sigma}(z_j)^{1/2} \mathbf{X}(z_j) + \boldsymbol{\mu}(z_j)^\top \mathbf{\Sigma}(z_i)^{1/2} \mathbf{X}(z_i) \\
&+ \sigma \mathbf{E}_i^\top \mathbf{\Sigma}(z_j)^{1/2} \mathbf{X}(z_j) + \sigma \mathbf{E}_j^\top \mathbf{\Sigma}(z_i)^{1/2} \mathbf{X}(z_i) \\
&+ \sigma \mathbf{E}_i^\top \boldsymbol{\mu}(z_j) + \sigma \mathbf{E}_j^\top \boldsymbol{\mu}(z_i) \\
&+ \sigma^2 (\mathbf{E}_i \cdot \mathbf{E}_j - \mathbb{E}[\mathbf{E}_i \cdot \mathbf{E}_j]).
\end{aligned}
$$

Using this decomposition, and a union bound to produce the result in terms of the maximum difference over all $i, j \in [n]$, we get

$$
\begin{aligned}
\pi_{\max} = \mathbb{P}\left(\max_{i,j \in [n]} |\mathbf{Y}_i \cdot \mathbf{Y}_j - \mathbb{E}[\mathbf{Y}_i \cdot \mathbf{Y}_j]| > tC_{ij}\right) &\leq \sum_{ij} \pi_{ij}(t) \\
&\leq \sum_{ij} \Bigg[ \mathbb{P}\left(\left|\mathbf{X}(z_i)^\top \mathbf{\Sigma}_{ij} \mathbf{X}(z_j) - \mathbb{E}[\mathbf{X}(z_i)^\top \mathbf{\Sigma}_{ij} \mathbf{X}(z_j)]\right| > \frac{t}{8} C_{ij}\right) && \text{(18)}
\end{aligned}
$$

$$
+ \mathbb{P}\left(\left|\boldsymbol{\mu}(z_i)^\top \mathbf{\Sigma}(z_j)^{1/2} \mathbf{X}(z_j)\right| > \frac{t}{8} C_{ij}\right) + \mathbb{P}\left(\left|\boldsymbol{\mu}(z_j)^\top \mathbf{\Sigma}(z_i)^{1/2} \mathbf{X}(z_i)\right| > \frac{t}{8} C_{ij}\right) \quad \text{(19)}
$$

$$
+ \mathbb{P}\left(\left|\sigma \mathbf{E}_i^\top \mathbf{\Sigma}(z_j)^{1/2} \mathbf{X}(z_j)\right| > \frac{t}{8} C_{ij}\right) + \mathbb{P}\left(\left|\sigma \mathbf{E}_j^\top \mathbf{\Sigma}(z_i)^{1/2} \mathbf{X}(z_i)\right| > \frac{t}{8} C_{ij}\right) \quad \text{(20)}
$$

$$
+ \mathbb{P}\left(\left|\sigma \mathbf{E}_i^\top \boldsymbol{\mu}(z_j)\right| > \frac{t}{8} C_{ij}\right) + \mathbb{P}\left(\left|\sigma \mathbf{E}_j^\top \boldsymbol{\mu}(z_i)\right| > \frac{t}{8} C_{ij}\right) \quad \text{(21)}
$$

$$
+ \mathbb{P}\left(\left|\sigma^2 (\mathbf{E}_i \cdot \mathbf{E}_j - \mathbb{E}[\mathbf{E}_i \cdot \mathbf{E}_j])\right| > \frac{t}{8} C_{ij}\right) \Bigg]. \quad \text{(22)}
$$

Next we use lemmas D.1-D.5 to bound the terms in expressions (18)-(22), respectively. Letting $c \geq c_1, \ldots, c_8$ and $n_4 \geq n_0, \ldots, n_3$, we get that for $n \geq n_4$:

$$
\begin{aligned}
\pi_{\max} &\leq \sum_{ij} \left[ 8 \exp\left(-(\log n)\frac{cM}{K^2} \min\left\{\frac{M}{K^2}, c\right\}\right) + 8 \exp\left(-\frac{cM^2}{K^2}(\log n)\right) \right] \\
&= 8 \exp\left(-(\log n)\frac{cM}{K^2} \min\left\{\frac{M}{K^2}, c\right\} + 2\log n\right) + 8\exp\left(-\frac{cM^2}{K^2}(\log n) + 2\log n\right).
\end{aligned}
$$

Now fix any $\varepsilon > 0$. By choosing $M$ large enough so that $\frac{M}{K^2} > c$, and then increasing $M$ if necessary, we can achieve

$$
\pi_{\max} \leq 8\exp\left(-\log n\left(\frac{c^2 M}{K^2} - 2\right)\right) + 8\exp\left(-\log n\left(\frac{cM^2}{K^2} - 2\right)\right) \leq \varepsilon,
$$

and the result of the theorem follows. $\qquad \square$

*Proof of Proposition 3.2.* We have

$$\mathbb{E}[\mathbf{Y}_i \cdot \mathbf{Y}_j] = \phi(z_i) \cdot \phi(z_j) + p\sigma^2 \mathbf{I}[i = j],$$

hence

$$
\left| \frac{\mathbf{Y}_i \cdot \mathbf{Y}_j}{p} - \frac{\phi(z_i) \cdot \phi(z_j)}{p} - \sigma^2 \mathbf{I}[i = j] \right|
$$

$$
= \left| \frac{\mathbf{Y}_i \cdot \mathbf{Y}_j}{\mathbb{E}[\|\mathbf{Y}_i\|^2]^{1/2} \mathbb{E}[\|\mathbf{Y}_j\|^2]^{1/2}} - \frac{\mathbb{E}[\mathbf{Y}_i \cdot \mathbf{Y}_j]}{\mathbb{E}[\|\mathbf{Y}_i\|^2]^{1/2} \mathbb{E}[\|\mathbf{Y}_j\|^2]^{1/2}} \right| \frac{\mathbb{E}[\|\mathbf{Y}_i\|^2]^{1/2} \mathbb{E}[\|\mathbf{Y}_j\|^2]^{1/2}}{p}
$$

$$
\leq \left| \frac{\mathbf{Y}_i \cdot \mathbf{Y}_j}{\mathbb{E}[\|\mathbf{Y}_i\|^2]^{1/2} \mathbb{E}[\|\mathbf{Y}_j\|^2]^{1/2}} - \frac{\mathbb{E}[\mathbf{Y}_i \cdot \mathbf{Y}_j]}{\mathbb{E}[\|\mathbf{Y}_i\|^2]^{1/2} \mathbb{E}[\|\mathbf{Y}_j\|^2]^{1/2}} \right| \frac{1}{p} \max_{i \in [n]} \mathbb{E}[\|\mathbf{Y}_i\|^2]
$$

where

$$
\frac{1}{p} \max_{i \in [n]} \mathbb{E}[\|\mathbf{Y}_i\|^2] = \max_{i \in [n]} \frac{\mathrm{tr}[\boldsymbol{\Sigma}(z_i)] + \|\boldsymbol{\mu}(z_i)\|^2}{p} + \sigma^2.
$$

The first claim of the proposition then follows from Theorem 3.1.

For the second claim of the proposition, note that for $i \neq j$,

$$
\frac{\mathsf{CosSim}(\phi(z_i), \phi(z_j))}{\gamma_{ij}(\sigma)}
$$

$$
= \frac{\phi(z_i) \cdot \phi(z_j)}{(\|\phi(z_i)\|^2 + p\sigma^2)^{1/2}(\|\phi(z_j)\|^2 + p\sigma^2)^{1/2}} = \frac{\mathbb{E}[\mathbf{Y}_i \cdot \mathbf{Y}_j]}{\mathbb{E}[\|\mathbf{Y}_i\|^2]^{1/2} \mathbb{E}[\|\mathbf{Y}_j\|^2]^{1/2}}.
$$

So we need to prove that

$$
\max_{i \neq j \in [n]} \left| \frac{\mathbf{Y}_i \cdot \mathbf{Y}_j}{\|\mathbf{Y}_i\| \|\mathbf{Y}_j\|} - \frac{\mathbb{E}[\mathbf{Y}_i \cdot \mathbf{Y}_j]}{\mathbb{E}[\|\mathbf{Y}_i\|^2]^{1/2} \mathbb{E}[\|\mathbf{Y}_j\|^2]^{1/2}} \right| \in O_{\mathbb{P}} \left( \sqrt{\frac{\log n}{\min_{i \in [n]} p_{\mathrm{eff}}^{(i)}}} \right). \tag{23}
$$

The only differences between this and the result of Theorem 3.1 is that here we only need to consider the maximum over $i \neq j$, and the normalisation by $\|\mathbf{Y}_i\| \|\mathbf{Y}_j\|$ of the first term within the absolute value in (23).

Consider the decomposition:

$$
\frac{\mathbf{Y}_i \cdot \mathbf{Y}_j}{\|\mathbf{Y}_i\| \|\mathbf{Y}_j\|} - \frac{\mathbb{E}[\mathbf{Y}_i \cdot \mathbf{Y}_j]}{\mathbb{E}[\|\mathbf{Y}_i\|^2]^{1/2} \mathbb{E}[\|\mathbf{Y}_j\|^2]^{1/2}}
$$

$$
= \frac{\mathbf{Y}_i \cdot \mathbf{Y}_j}{\mathbb{E}[\|\mathbf{Y}_i\|^2]^{1/2} \mathbb{E}[\|\mathbf{Y}_j\|^2]^{1/2}} - \frac{\mathbb{E}[\mathbf{Y}_i \cdot \mathbf{Y}_j]}{\mathbb{E}[\|\mathbf{Y}_i\|^2]^{1/2} \mathbb{E}[\|\mathbf{Y}_j\|^2]^{1/2}} \tag{24}
$$

$$
+ \mathbf{Y}_i \cdot \mathbf{Y}_j \left[ \frac{1}{\|\mathbf{Y}_i\| \|\mathbf{Y}_j\|} - \frac{1}{\mathbb{E}[\|\mathbf{Y}_i\|^2]^{1/2} \mathbb{E}[\|\mathbf{Y}_j\|^2]^{1/2}} \right]. \tag{25}
$$

Theorem 3.1 implies that the maximum over $i \neq j$ of the term in (24) is in $O_{\mathbb{P}} \left( \sqrt{\frac{\log n}{\min_{i \in [n]} p_{\mathrm{eff}}^{(i)}}} \right)$.

Consider the term in (25). Using the triangle inequality, Cauchy-Schwarz, and triangle inequality again, we have:

$$
\left| \mathbf{Y}_i \cdot \mathbf{Y}_j \left[ \frac{1}{\|\mathbf{Y}_i\| \|\mathbf{Y}_j\|} - \frac{1}{\mathbb{E}[\|\mathbf{Y}_i\|^2]^{1/2} \mathbb{E}[\|\mathbf{Y}_j\|^2]^{1/2}} \right] \right|
$$

$$
= \left| \frac{\mathbf{Y}_i}{\|\mathbf{Y}_i\|} \cdot \left( \frac{\mathbf{Y}_j}{\|\mathbf{Y}_j\|} - \frac{\mathbf{Y}_j}{\mathbb{E}[\|\mathbf{Y}_j\|^2]^{1/2}} \right) + \frac{\mathbf{Y}_j}{\mathbb{E}[\|\mathbf{Y}_j\|^2]^{1/2}} \cdot \left( \frac{\mathbf{Y}_i}{\|\mathbf{Y}_i\|} - \frac{\mathbf{Y}_i}{\mathbb{E}[\|\mathbf{Y}_i\|^2]^{1/2}} \right) \right|
$$

$$
\leq \|\mathbf{Y}_j\| \left| \frac{1}{\|\mathbf{Y}_j\|} - \frac{1}{\mathbb{E}[\|\mathbf{Y}_j\|^2]^{1/2}} \right| + \frac{\|\mathbf{Y}_j\|}{\mathbb{E}[\|\mathbf{Y}_j\|^2]^{1/2}} \|\mathbf{Y}_i\| \left| \frac{1}{\|\mathbf{Y}_i\|} - \frac{1}{\mathbb{E}[\|\mathbf{Y}_i\|^2]^{1/2}} \right|
$$

$$
\leq \left| \frac{\|\mathbf{Y}_j\|}{\mathbb{E}[\|\mathbf{Y}_j\|^2]^{1/2}} - 1 \right| + \left| \frac{\|\mathbf{Y}_j\|}{\mathbb{E}[\|\mathbf{Y}_j\|^2]^{1/2}} - 1 \right| \left| \frac{\|\mathbf{Y}_i\|}{\mathbb{E}[\|\mathbf{Y}_i\|^2]^{1/2}} - 1 \right| + \left| \frac{\|\mathbf{Y}_i\|}{\mathbb{E}[\|\mathbf{Y}_i\|^2]^{1/2}} - 1 \right|. \tag{26}
$$

Theorem 3.1 implies

$$
\max_i \left| \frac{\|\mathbf{Y}_i\|^2}{\mathbb{E}[\|\mathbf{Y}_i\|^2]} - 1 \right| \in O_{\mathbb{P}} \left( \sqrt{\frac{\log n}{\min_i p_{\mathrm{eff}}^{(i)}}} \right), \tag{27}
$$

i.e., for any $\epsilon > 0$ there exists $M$ and $n_0$ such that for any $n \geq n_0$,

$$\mathbb{P}\left(\max_i \left|\frac{\|\mathbf{Y}_i\|}{\mathbb{E}[\|\mathbf{Y}_i\|^2]^{1/2}} - 1\right| > M\sqrt{\frac{\log n}{\min_i p_{\text{eff}}^{(i)}}}\right) \leq \epsilon.$$

Now for any $a > 0$, $(a+1)|a-1| = |a^2 - 1|$, i.e. $|a-1| \leq |a^2 - 1|$, hence

$$\max_i \left|\frac{\|\mathbf{Y}_i\|}{\mathbb{E}[\|\mathbf{Y}_i\|^2]^{1/2}} - 1\right| > M\sqrt{\frac{\log n}{\min_i p_{\text{eff}}^{(i)}}} \quad \Rightarrow \quad \max_i \left|\frac{\|\mathbf{Y}_i\|^2}{\mathbb{E}[\|\mathbf{Y}_i\|^2]} - 1\right| > M\sqrt{\frac{\log n}{\min_i p_{\text{eff}}^{(i)}}},$$

so (27) implies:

$$\max_i \left|\frac{\|\mathbf{Y}_i\|}{\mathbb{E}[\|\mathbf{Y}_i\|^2]^{1/2}} - 1\right| \in O_{\mathbb{P}}\left(\sqrt{\frac{\log n}{\min_i p_{\text{eff}}^{(i)}}}\right).$$

Thus, via (26), the maximum over $i \neq j$ of the term in (25) is in $O_{\mathbb{P}}\left(\sqrt{\frac{\log n}{\min_i p_{\text{eff}}^{(i)}}}\right)$. We have thus shown that (23) holds as

required, and that completes the proof of the second claim of the proposition. $\qquad\square$

*Proof of Lemma 3.3.* We have:

$$\begin{aligned}
\mathbb{E}[\|\mathbf{Y}^{\text{nf}}(z) - \mathbf{Y}^{\text{nf}}(z')\|^2] &= \mathbb{E}[\|\mathbf{Y}^{\text{nf}}(z)\|^2] + \mathbb{E}[\|\mathbf{Y}^{\text{nf}}(z')\|^2] - 2\mathbb{E}[\mathbf{Y}^{\text{nf}}(z) \cdot \mathbf{Y}^{\text{nf}}(z')] \\
&= \|\phi(z)\|^2 + \|\phi(z')\|^2 - 2\phi(z) \cdot \phi(z') \\
&= \|\phi(z) - \phi(z')\|^2.
\end{aligned}$$

Therefore assumption **A3** is equivalent to continuity of $\phi$, and **A5** implies $\phi$ is invertible on its image $\mathcal{M}$. A result in the theory of metric spaces [Prop. 13.26](Sutherland, 2009), states that any continuous invertible mapping on a compact domain has an continuous inverse. Therefore the inverse of $\phi$ is continuous. $\qquad\square$

## E. Supporting material for Section 4

### E.1. Bounding the bottleneck distance

First let us collect the following definitions and results from Chazal et al. (2013, Section 2.1):

1. The Hausdorff distance between two compact subsets $C_1$ and $C_2$ of a metric space is

$$d_{\text{H}}(C_1, C_2) = \max\{\sup_{x \in C_1} d(x, C_2), \sup_{x \in C_2} d(x, C_1)\},$$

where the distance $d(x, C)$ to a compact subset $C$ is the minimum over all $y \in C$ of $d(x, y)$.

2. Two compact metric spaces are said to be isometric if there exists a bijection between them that preserves distances.

3. The Gromov-Hausdorff distance $d_{\text{GH}}(\mathcal{M}, \mathcal{M}')$ between two compact metric spaces $\mathcal{M}$ and $\mathcal{M}'$ is the infimum of the real numbers $r \geq 0$ such that there exists a metric space with compact subspaces $C_1$ and $C_2$ which are isometric to $\mathcal{M}$ and $\mathcal{M}'$ respectively and such that $d_{\text{H}}(C_1, C_2) < r$. If $\mathcal{M}$ and $\mathcal{M}'$ are compact subsets of the same metric space, then $d_{\text{GH}}(\mathcal{M}, \mathcal{M}') \leq d_{\text{H}}(\mathcal{M}, \mathcal{M}')$.

4. Let $\text{Filt}(\mathcal{M})$ denote either the Rips, Čech, or Alpha filtration of $\mathcal{M}$, $\text{dgm}(\text{Filt}(\mathcal{M}))$ the associated persistent diagram (simply writing $\text{dgm}(\mathcal{M})$ for brevity in the main document), and $d_{\text{b}}$ the bottleneck distance between persistence diagrams. Then, for two compact metric spaces $\mathcal{M}, \mathcal{M}'$,

$$d_{\text{b}}\left(\text{dgm}(\text{Filt}(\mathcal{M})), \text{dgm}(\text{Filt}(\mathcal{M}'))\right) \leq 2d_{\text{GH}}(\mathcal{M}, \mathcal{M}').$$

Applying the above,

$$\begin{aligned}
d_{\text{b}}\left(\text{dgm}(\text{Filt}(\mathcal{Y}_n)), \text{dgm}(\text{Filt}(\mathcal{M}))\right) &\leq 2d_{\text{GH}}(\mathcal{Y}_n, \mathcal{M}) \leq 2(d_{\text{GH}}(\mathcal{Y}_n, \mathcal{M}_n) + d_{\text{GH}}(\mathcal{M}_n, \mathcal{M})) \\
&\leq 2(d_{\text{GH}}(\mathcal{Y}_n, \mathcal{M}_n) + d_{\text{H}}(\mathcal{M}_n, \mathcal{M}))
\end{aligned}$$

### E.2. Bound on the Gromov-Hausdorff distance

Here, we prove that

$$d^2_{\text{GH}}(p^{-1/2}\mathcal{Y}_n, p^{-1/2}\mathcal{M}_n) \leq \max_{i,j\in[n]} \frac{1}{p} |\mathbf{Y}_i \cdot \mathbf{Y}_j - \phi(z_i) \cdot \phi(z_j)| .$$

Let $\mathbf{W}_i := \phi(z_i)$. Then,

$$\begin{aligned}
&\left| \|\mathbf{Y}_i - \mathbf{Y}_j\|^2 - \|\mathbf{W}_i - \mathbf{W}_j\|^2 \right| \\
&= |(\mathbf{Y}_i \cdot \mathbf{Y}_i - \mathbf{W}_i \cdot \mathbf{W}_i) + (\mathbf{Y}_j \cdot \mathbf{Y}_j - \mathbf{W}_j \cdot \mathbf{W}_j) - 2(\mathbf{Y}_i \cdot \mathbf{Y}_j - \mathbf{W}_i \cdot \mathbf{W}_j)| \\
&\leq 4\epsilon.
\end{aligned}$$

where $\epsilon := \max_{i,j\in[n]} |\mathbf{Y}_i \cdot \mathbf{Y}_j - \mathbf{W}_i \cdot \mathbf{W}_j| .$

Applying e.g. Whiteley et al. (2021, Lemma 6), we have $||a| - |b|| \leq |a^2 - b^2|^{1/2}$ for any $a, b \in \mathbb{R}$, therefore,

$$\left| \|\mathbf{Y}_i - \mathbf{Y}_j\| - \|\mathbf{W}_i - \mathbf{W}_j\| \right| \leq \sqrt{4\epsilon}, \tag{28}$$

for all $i, j \in [n]$.

Now, let us recall some definitions and results from Ivanov et al. (2016):

1. A *correspondence* between two metric spaces $(\mathcal{M}, d)$ and $(\mathcal{M}', d')$ is a subset $R \subset \mathcal{M} \times \mathcal{M}'$ such that the canonical projections $\pi_1 : (x, y) \mapsto x$ and $\pi_1 : (x, y) \mapsto y$ for $(x, y) \in R$ are surjective, and the set of all correspondences is denoted $\mathcal{R}(\mathcal{M}, \mathcal{M}')$.

2. The distortion of a correspondence is

$$\text{dis}(R) := \sup \{|d(x, x') - d'(y, y')| : (x, y), (x', y') \in R\},$$

3. The Gromov-Hausdorff distance $d_{\text{GH}}(\mathcal{M}, \mathcal{M}')$ quantifies the 'best correspondence' (Ivanov et al., 2016, Theorem 1.1),

$$d_{\text{GH}}(\mathcal{M}, \mathcal{M}') = \frac{1}{2} \inf\{\text{dis}(R) : R \in \mathcal{R}(\mathcal{M}, \mathcal{M}')\}.$$

The set $R_0 = \{(\mathbf{Y}_1, \mathbf{W}_1), \ldots, (\mathbf{Y}_n, \mathbf{W}_n)\}$ is a correspondence between $\mathcal{Y}_n$ and $\mathcal{M}_n$, therefore,

$$d_{\text{GH}}(\mathcal{Y}_n, \mathcal{M}_n) = \frac{1}{2} \inf\{\text{dis}(R) : R \in \mathcal{R}(\mathcal{Y}_n, \mathcal{M}_n) \leq \frac{1}{2}\text{dis}(R_0) \leq \frac{1}{2}\sqrt{4\epsilon} = \sqrt{\epsilon},$$

where the penultimate inequality uses (28). Therefore,

$$\begin{aligned}
d^2_{\text{GH}}(p^{-1/2}\mathcal{Y}_n, p^{-1/2}\mathcal{M}_n) &= \frac{1}{p}d^2_{\text{GH}}(\mathcal{Y}_n, \mathcal{M}_n) \leq \frac{1}{p}\epsilon \\
&= \max_{i,j\in[n]} \frac{1}{p} |\mathbf{Y}_i \cdot \mathbf{Y}_j - \phi(z_i) \cdot \phi(z_j)| \leq \max_{i,j\in[n]} \frac{1}{p} \left|\mathbf{Y}_i \cdot \mathbf{Y}_j - \phi(z_i) \cdot \phi(z_j) - \sigma^2\mathbf{I}[i = j]\right| + \sigma^2
\end{aligned}$$

,

as claimed at the end of Section 4.

## F. TDA and manifold learning using normalised data $\mathbf{Y}_i / \|\mathbf{Y}_i\|$

In connection with the second part of Proposition 3.2, we can consider TDA and manifold learning based on the 'self-normalised' data,

$$\begin{aligned}
\overline{\mathcal{Y}}_n &:= \{\mathbf{Y}_i / \|\mathbf{Y}_i\|; i \in [n]\} \\
\overline{\mathcal{M}}_n &:= \{\phi(z_i) / \|\phi(z_i)\|; i \in [n]\}, \\
\overline{\mathcal{M}} &:= \{\phi(z) / \|\phi(z)\|; z \in \mathcal{Z}\}.
\end{aligned}$$

A practical advantage of this 'self-normalised' setting is that we do not need to decide on a specific rescaling factor such as $p$ in the first part of Proposition 3.2.

We can consider the question of how $\overline{\mathcal{M}}$ relates to $\mathcal{Z}$, analogously to Lemma 3.3 but under the following assumption instead of **A5**.

**A6.** If $z \neq z'$, then:

$$\mathbb{E}\left[\left\|\frac{\mathbf{Y}^{\mathrm{nf}}(z)}{\mathbb{E}[\|\mathbf{Y}^{\mathrm{nf}}(z)\|^2]^{1/2}} - \frac{\mathbf{Y}^{\mathrm{nf}}(z')}{\mathbb{E}[\|\mathbf{Y}^{\mathrm{nf}}(z')\|^2]^{1/2}}\right\|^2\right] > 0.$$

**Lemma F.1.** *Under A3, A4 and A6, $z \mapsto \phi(z)/\|\phi(z)\|$ is a homeomorphism between $\mathcal{Z}$ and $\overline{\mathcal{M}}$.*

*Proof.* Using (5),

$$\begin{aligned}
\frac{1}{2}\left\|\frac{\phi(z)}{\|\phi(z)\|} - \frac{\phi(z')}{\|\phi(z')\|}\right\|^2 &= 1 - \frac{\phi(z) \cdot \phi(z')}{\|\phi(z)\|\|\phi(z')\|} \\
&= 1 - \frac{\mathbb{E}[\mathbf{Y}^{\mathrm{nf}}(z) \cdot \mathbf{Y}^{\mathrm{nf}}(z')]}{\mathbb{E}[\|\mathbf{Y}^{\mathrm{nf}}(z)\|^2]^{1/2}\mathbb{E}[\|\mathbf{Y}^{\mathrm{nf}}(z')\|^2]^{1/2}} \\
&= \frac{1}{2}\mathbb{E}\left[\left\|\frac{\mathbf{Y}^{\mathrm{nf}}(z)}{\mathbb{E}[\|\mathbf{Y}^{\mathrm{nf}}(z)\|^2]^{1/2}} - \frac{\mathbf{Y}^{\mathrm{nf}}(z')}{\mathbb{E}[\|\mathbf{Y}^{\mathrm{nf}}(z')\|^2]^{1/2}}\right\|^2\right].
\end{aligned}$$

The claim of the lemma then follows by the same arguments as in the proof Lemma 3.3. □

Lemma F.1 motivates the application of TDA to $\overline{\mathcal{Y}}_n$ to discover the structure of $\mathcal{Z}$. Following the same line of investigation as in Section E.2, in order to bound $d_{\mathrm{GH}}^2(\overline{\mathcal{Y}}_n, \overline{\mathcal{M}}_n)$ we can follow similar arguments as above subject to some small changes: let $\mathbf{W}_i := \phi(z_i)/\|\phi(z_i)\|$, so $\|\mathbf{W}_i\| = 1$, and

$$\begin{aligned}
&\left|\left\|\frac{\mathbf{Y}_i}{\|\mathbf{Y}_i\|} - \frac{\mathbf{Y}_j}{\|\mathbf{Y}_j\|}\right\|^2 - \|\mathbf{W}_i - \mathbf{W}_j\|^2\right| \\
&= \left|\left(\frac{\mathbf{Y}_i}{\|\mathbf{Y}_i\|} \cdot \frac{\mathbf{Y}_i}{\|\mathbf{Y}_i\|} - \mathbf{W}_i \cdot \mathbf{W}_i\right) + \left(\frac{\mathbf{Y}_j}{\|\mathbf{Y}_j\|} \cdot \frac{\mathbf{Y}_j}{\|\mathbf{Y}_j\|} - \mathbf{W}_j \cdot \mathbf{W}_j\right) - 2\left(\frac{\mathbf{Y}_i}{\|\mathbf{Y}_i\|} \cdot \frac{\mathbf{Y}_j}{\|\mathbf{Y}_j\|} - \mathbf{W}_i \cdot \mathbf{W}_j\right)\right| \\
&= \left|0 + 0 + 2\left|\frac{\mathbf{Y}_i}{\|\mathbf{Y}_i\|} \cdot \frac{\mathbf{Y}_j}{\|\mathbf{Y}_j\|} - \mathbf{W}_i \cdot \mathbf{W}_j\right|\right. \\
&\leq 2\epsilon,
\end{aligned}$$

where $\epsilon := \max_{i \neq j \in [n]}\left|\frac{\mathbf{Y}_i}{\|\mathbf{Y}_i\|} \cdot \frac{\mathbf{Y}_j}{\|\mathbf{Y}_j\|} - \mathbf{W}_i \cdot \mathbf{W}_j\right|$. Following through the same steps as in Section E.2 then gives:

$$d_{\mathrm{GH}}^2(\overline{\mathcal{Y}}_n, \overline{\mathcal{M}}_n) \leq \max_{i \neq j}\left|\frac{\mathbf{Y}_i \cdot \mathbf{Y}_j}{\|\mathbf{Y}_i\|\|\mathbf{Y}_j\|} - \frac{\phi(z_i) \cdot \phi(z_j)}{\|\phi(z_i)\|\|\phi(z_j)\|}\right|.$$

Applying the triangle inequality and adopting the notation of the second part of Proposition 3.2 we therefore have:

$$d_{\mathrm{GH}}^2(\overline{\mathcal{Y}}_n, \overline{\mathcal{M}}_n) \leq \max_{i \neq j}\left|\mathsf{CosSim}(\mathbf{Y}_i, \mathbf{Y}_j) - \frac{\mathsf{CosSim}(\phi(z_i), \phi(z_j))}{\gamma_{ij}(\sigma)}\right| + 1 - \frac{1}{\max_{i \neq j} \gamma_{ij}(\sigma)},$$

where Cauchy-Schwarz and $\gamma_{ij}(\sigma) \geq 1$ have been used. The first $\max_{i \neq j}$ term on the l.h.s. of the above inequality is controlled by Proposition 3.2, where as the second term goes to zero as $\sigma \to 0$.

The task of looking for evidence of isometry of the mapping $z \mapsto \phi(z)/\|\phi(z)\|$ can be conducted as in Section 5, simply replacing $\mathbf{Y}_i$ there by $\mathbf{Y}_i/\|\mathbf{Y}_i\|$; thus comparing shortest path-lengths in $\overline{\mathcal{Y}}_n$ to those in $\mathcal{Z}_n$.

## G. Further discussion of the toy example

Recall in the toy example:

$$\phi(z) = p^{1/2}\left[z_1, \frac{2}{\pi}\sin(\pi z_2/2), \frac{2}{\pi}\cos(\pi z_2/2)\right],$$

with $z = (z_1, z_2) \in \{(z_1, z_2) : z_1^2 + z_2^2\} \subset \mathbb{R}^2$. We have:

$$\frac{1}{p}\phi(z)\cdot\phi(z') = z_1 z_1' + \frac{4}{\pi^2}\sin(z_2\pi/2)\sin(z_2'\pi/2) + \frac{4}{\pi^2}\cos(z_2\pi/2)\cos(z_2'\pi/2).$$

$$= z_1 z_1' + \frac{4}{\pi^2}\cos[(z_2 - z_2')\pi/2].$$

It can be shown that isometry holds, indeed by direct calculation:

$$\left.\frac{\partial^2}{\partial z_i \partial z_j'}\phi(z)\cdot\phi(z')\right|_{z=z'} = \begin{cases} p, & i = j, \\ 0, & i \neq j, \end{cases}$$

which according to (Whiteley et al., 2021, Thm. 1) is a sufficient condition for isometry (the above expression for the partial derivatives means the matrix denoted by $\mathbf{H}_{\eta_t}$ in (Whiteley et al., 2021, Thm. 1) is proportional to the identity matrix for all $t$).

The visualisation in Figure 1 was produced as follows. Write $\mathbf{Y} \coloneqq [\mathbf{Y}_1|\cdots|\mathbf{Y}_n]^\top$ and consider the SVD:

$$\mathbf{Y} = \mathbf{U}_3\mathbf{S}_3\mathbf{V}_3^\top + \mathbf{U}_\perp\mathbf{S}_\perp\mathbf{V}_\perp^\top,$$

where $\mathbf{S}_3$ is the 3x3 diagonal matrix with diagonal elements which are the three largest singular values, and the columns of $\mathbf{U}_3 \in \mathbb{R}^{n\times 3}$, $\mathbf{V}_3 \in \mathbb{R}^{p\times 3}$ are associated left/right singular vectors.

Subplots (a), (b), (c) in Figure 1 show for respective values of $p$, the rows of $\mathbf{U}_3\mathbf{S}_3$ as a point-cloud in $\mathbb{R}^3$, rotated to align with (d) (this alignment accounts for the fact that singular vectors are only mathematically defined up to sign, so the sign obtained when computing the SVD in practice is arbitrary).

## H. Supplementary Grid-cell Experiments

### H.1. Context

In this section, we will expand on the details necessary to reproduce the results in Section 6.3. Gardner et al. (2022) collected data over multiple days, from three different rats, covering various modules (groups of grid cells) within each rat's brain. In all of our experiments, we focus on a single dataset: {rat 'R', module 2, day 1}, although this choice is arbitrary.

In order to visualise the toroidal structure in the data, as in Figure 3(a), we follow the approach of Gardner et al. (2022): perform PCA on the data into six dimensions, followed by UMAP (McInnes et al., 2018) into three dimensions, and colour points by the first PC. We follow this procedure for $n = 15,000$ points, which were selected by retaining the 15,000 most active points, as measured by the mean firing rate in each time bin. The hyperparameters used for UMAP were: 'n_components'=3, 'n_neighbors'=2000, 'min_dist'=0.8, 'metric'='cosine' and 'init'='spectral'. Running the UMAP algorithm on 15,000 points had the longest running time of any of our experiments, taking 3 minutes 32 seconds. This was run on a Linux-based Dell Latitude laptop equipped with an 11th Gen Intel Core i7-1185G7 CPU (4 cores, 8 threads) and 15GiB of RAM, using Python v3.11.

We also replicate the persistent homology analysis in Gardner et al. (2022), using the `ripser` Python package. Since computing the persistent homology of a set of points is computationally expensive and sensitive to outliers, it is common to downsample and dimension-reduce the data beforehand. Following the approach used for the torus visualisation, we begin by selecting the same 15,000 data points and apply PCA into six dimensions. The choice of six dimensions is explored in detail in Gardner et al. (2022), where they show that the first 6 dimensions retain a large proportion of the variance in the data. Using the authors' code (Gardner et al., 2021a), we then downsample 800 points from the PCA embedding using a density-based method informed by the fuzzy topological representation used internally by UMAP. Finally, a distance matrix is produced from this reduced point cloud, which can be passed to `ripser`. Further details of this procedure can be found in Gardner et al. (2022). The output of `ripser` can easily be plotted as a persistence diagram, as in Figure 3(c). We note

that for purposes of visualisation, it was necessary to slightly shift one of the points in persistence diagram corresponding to on of the 1-dimensional holes in order to visually distinguish it from the other, as their birth and death times were close to identical. We also note that Gardner et al. (2022) reported bar-code diagrams, rather than persistence diagrams, but these convey the same information; we show a persistence diagram to connect more directly with Section 4.

To produce plots (b) and (d) in Figure 3, we applied the cohomological decoding procedure introduced by (De Silva & Vejdemo-Johansson, 2009) and implemented in Gardner et al. (2021a). This method produces two circular coordinates, each corresponding to one of the two 1-dimensional holes identified in the persistent homology analysis. These coordinates, referred to as *toroidal coordinates* are plotted in Gardner et al. (2022) as functions of physical space to reveal the underlying periodic structure in the data. Gardner et al. (2022) show that the periodic structure can be represented by two vectors, $\mathbf{r}_1$ and $\mathbf{r}_2$, that together define a rhombus that, when tessellated, captures the repeated pattern in the toroidal coordinates. In (d), we simply plot the tessellated rhombus atop the physical locations (coloured by the first PC). In (b), the points inside each of the rhombi are identified and then superimposed onto the central rhombus. To help the reader visualise how this 2D rhombus relates to a 3D torus, we have added markings in red to illustrate that a torus is formed when opposite edges of the rhombus are "glued" together.

### H.2. Searching for isometry

This section contains additional analyses of the grid cell data of Gardner et al. (2022), expanding on the results presented in Section 6.3. While Gardner et al. (2022) established how circular coordinates on the inferred torus align with physical $x$-$y$ coordinates, we go a step further by asking a geometric question:

is there evidence of *isometry* between the toroidal structure found in $\mathbf{Y}_1, \ldots, \mathbf{Y}_n$ and physical space?

To frame this question, we consider the grid-cell firing data $\mathbf{Y}_1, \ldots, \mathbf{Y}_n$ to be generated from three alternative instances of the random function model from Section 3, each of which is built upon a different choice of metric space $(\mathcal{Z}, d_{\mathcal{Z}})$ used to represent physical locations. In each case, we then look for evidence of isometry between $\mathcal{Z}$ and $\mathcal{M}$.

**Model 1: Open field, with real-world Euclidean distance.** $\mathcal{Z}$ is the open field of all real-world physical locations the rat could have possibly occupied, specifically the square $[-0.75, 0.75]^2 \subset \mathbb{R}^2$ and $z_i = \xi_i$ is the physical location of the rat at the $i$th time point. The metric $d_{\mathcal{Z}}$ is Euclidean distance, i.e. straight-line distance. Making no use of the topological analysis in Gardner et al. (2022), this model can be regarded as the default hypothesis for how physical geometry relates to the geometry of $\mathcal{M}$ and hence $\mathcal{Y}_n$.

**Model 2: Superimposition on rhombus, with Euclidean distance.** $\mathcal{Z}$ is the central rhombus in the tesselation of $[-0.75, 0.75]^2$ obtained by Gardner et al. (2022) (see Figure 6 (b), left plot) and $z_i$ is the superimposition onto this rhombus of the physical location of the rat at the $i$th time point (Figure 6 (b), middle plot). The metric $d_{\mathcal{Z}}$ is Euclidean distance on the rhombus. Unlike Model 1, Model 2 accounts for the tessellated rhomboidal pattern in the firing data as a function of physical space.

**Model 3: Superimposition on rhombus, with Euclidean teleportation distance.** $\mathcal{Z}$ and $z_i$ are the same as in Model 2, but $d_{\mathcal{Z}}$ is Euclidean distance subject to periodic boundary conditions on the rhombus, i.e., points on opposing edges of the rhombus correspond to the same position (same circular coordinates) and therefore paths on $\mathcal{Z}$ can 'teleport' at the edges to the corresponding point on the opposite edge. Defining $d_{\mathcal{Z}}$ in this way extends Model 2 to respect toroidal topology.

In Models 2 and 3, the choice of superimposing on the central rhombus rather than any of the others is arbitrary – all that matters is the superimposition of physical locations on to one common rhombus. The difference between the three models is illustrated in Figure 6(b). Let $\xi_i, \xi_j$ be two physical locations which the rat occupied at some points $i, j$ in time, indicated by respectively blue and green dots in the left plot of Figure 6(b). Under Model 1, where $z_i = \xi_i$ and $z_j = \xi_j$, the shortest path in $\mathcal{Z}_n$ between these two points is shown in green. In Model 2, $z_i$ and $z_j$ are the superimposition of $\xi_i$ and $\xi_j$ onto the central rhombus; for the particular points in the example of 6(b) middle plot, we have $\xi_i = z_i$ (blue), but $\xi_j \neq z_j$ (green). The shortest path in $\mathcal{Z}_n$ is shown in green. In Model 3, $\xi_i$ and $\xi_j$ are represented by the same points $z_i$ and $z_j$ as in Model 2, but the definition of distance and hence shortest path is different because of 'teleporting', represented by a dashed line in the right plot of 6(b), is allowed.

In our experiments, we work with the normalised data vectors $\tilde{\mathbf{Y}}_i := \mathbf{Y}_i / \|\mathbf{Y}_i\|, i \in [n]$, where the $n = 15,000$ points are the same as those in our contextual experiments. From a theoretical perspective, this normalisation acts as a natural rescaling of the data to be applied by default as in the second part of Proposition 3.2; the first part of proposition 3.2 indicates that an

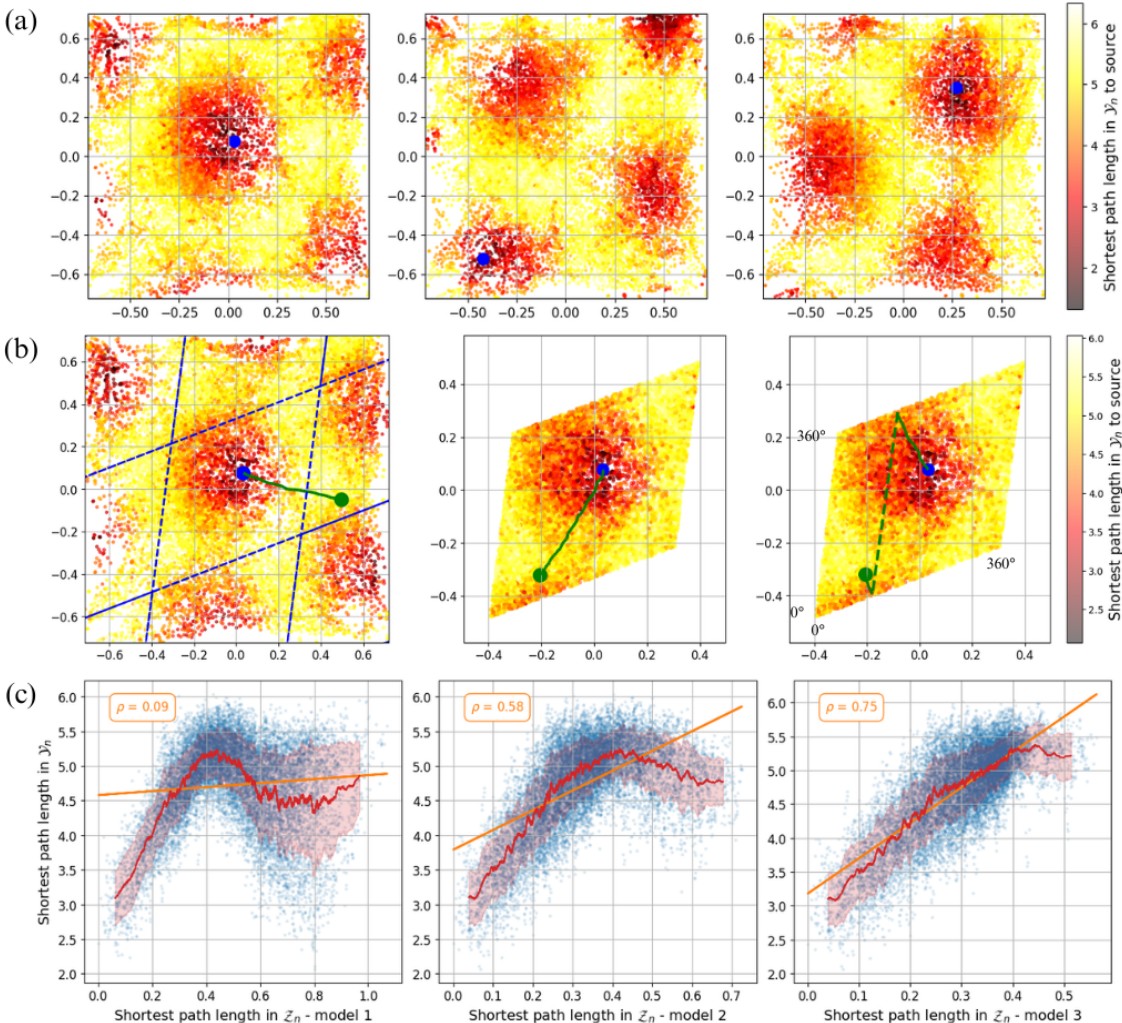

*Figure 6.* **(a)** For 3 possible source locations shown in blue, all other physical locations visited by the rat are colored by the shortest path-length in $\mathcal{Y}_n$ from the source. **(b)** *Left:* The tessellated rhombus is plotted atop the physical locations. Two physical locations (blue, green), and the shortest path in $\mathcal{Z}_n$ between them under Model 1 are shown. *Middle*: Representations of the same source and sink locations under Model 2, in which physical locations are superimposed on the central rhombus in the tesselation. Shortest path in $\mathcal{Z}_n$ under Model 2 is shown. *Right*: Under Model 3, the distance $d_{\mathcal{Z}}$ allows for 'teleporting' (dashed line). **(c)** relationships between shortest path-lengths in $\mathcal{Y}_n$ and in $\mathcal{Z}_n$ for Models 1-3 (left to right). Orange lines show best linear fit and red lines show moving averages (shading for $\pm$1s.d.). Strongest evidence of isometry appears for Model 3.

alternative scaling, $\mathbf{Y}_i/p^{1/2}$, would make sense if $p^{-1}\max_{i\in[n]}(\text{tr}[\boldsymbol{\Sigma}(z_i)] + \|\boldsymbol{\mu}(z_i)\|^2) + \sigma^2 \in O(1)$, but in practice we may not be sure that condition is satisfied.

As discussed in Section 5, we use a $k$-nearest neighbour graph (in our case $k = 10$), with edge weights $W_{ij} = \|\tilde{\mathbf{Y}}_i - \tilde{\mathbf{Y}}_j\|$, to approximate shortest path lengths in the point cloud $\mathcal{Y}_n := \{\tilde{\mathbf{Y}}_1, \ldots, \tilde{\mathbf{Y}}_n\}$. The shortest paths between points, denoted $\hat{L}(\mathbf{Y}_i, \mathbf{Y}_j)$, are calculated using Dijkstra's algorithm (Dijkstra, 1959) on the constructed graph. To aid visualisation of the pattern in the data over physical space, as illustrated in Figure 6(a), we smooth the calculated on-manifold path lengths as a weighted combination of the $k = 10$ nearest neighbours in physical space. Specifically, for a source position $\xi_i$ and some other physical position $\xi_j$, with neighbourhood $\mathcal{N}_j$, we weight each point $\xi_k \in \mathcal{N}_j$ by

$$w_k = \frac{\max_{l\in\mathcal{N}_j}(\|\xi_l - \xi_i\| - \|\xi_k - \xi_i\|)}{\sum_{k\in\mathcal{N}_j}\max_{l\in\mathcal{N}_j}(\|\xi_l - \xi_i\| - \|\xi_k - \xi_i\|)},$$

and set the smoothed path length to the source to be $\tilde{L}(\mathbf{Y}_i, \mathbf{Y}_j) := \sum_{k\in\mathcal{N}_j} w_k \cdot \hat{L}((\mathbf{Y}_i, \mathbf{Y}_k))$. These smoothed path lengths are also used in figures (b) and (c).

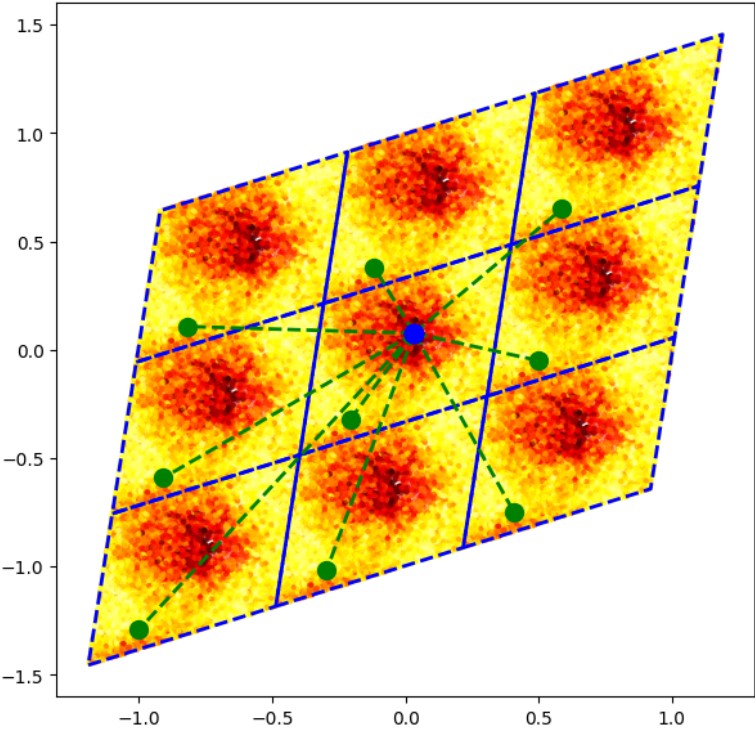

*Figure 7.* Shortest path lengths on $\mathcal{Z}_n$ from Model 3 are calculated by first re-tessellating the $z_i$ from Model 2 into the 8 rhombi surrounding the central rhombus (illustrated above). Shortest paths to the same destination point in each of the rhombi are calculated and the minimum of the 9 distances is taken to be the distance under Model 3. Here, the shortest path is to the point in the rhombus directly above the central rhombus.

The shortest path lengths in the open field, i.e. under Model 1, are calculated similarly: using a $k = 10$ nearest neighbour graph constructed from the physical positions $z_i = \xi_i$, followed by Dijkstra's algorithm. For Model 2, the procedure is the same, except the positions $z_i$ are now superimposed onto the central rhombus. Model 3 introduces additional complexity, as paths are allowed to teleport at the edges to the corresponding point on the opposite side. We handle this by 're-tessellating' the superimposed $z_i$ from Model 2 to the 8 surrounding rhombi, yielding a total of $135,000 = 9 \times 15,000$ points. Each $z_i$ now has 9 corresponding locations: one in each of the rhombi. Keeping the source point fixed in the central rhombus, we compute the shortest path lengths to each of the 9 destination points. These paths are illustrated in Figure 7. The minimum of these 9 distances is taken to be the path length under Model 3; in this case, the shortest path is to the point in the rhombus directly above the central rhombus.

On examination of the three distance-distance plots in Figure 6(c), we see distinct relationships between distances on the manifold $\mathcal{Y}_n$ and distances under models 1-3. Under Model 1, the relationship appears approximately linear for points near the source (within a distance of 0.4 on $\mathcal{Z}_n$). Beyond this threshold, however, distances on $\mathcal{Y}_n$ begin to decrease as distances on $\mathcal{Z}_n$ increase, with the trend reversing again around a distance of 0.8 on $\mathcal{Z}_n$. This relationship can be intuitively explained by the toroidal structure in $\mathcal{Y}_n$: as the rat moves in a straight line through physical space, its corresponding position on $\mathcal{Y}_n$ wraps around the torus, eventually approaching its starting position from the opposite direction.

Under Model 2, superimposing all points onto the central rhombus essentially maps the entire physical space onto a single, flat torus (a torus whose surface has zero Gaussian curvature everywhere). This accounts for the repeating spacial pattern observed in the data. However, flattening the torus into a 2D shape introduces two 'cuts' - effectively "slicing" the torus - which restricts the available paths. As a result, some shortest paths on the torus become inaccessible after slicing, forcing detours that increase the measured distance. This constraint explains the deviation from a linear trend for distances greater than around 0.4 on $\mathcal{Z}_n$. Model 3 addresses this limitation by employing our 're-tessellation' procedure, which allows paths to traverse the full toroidal structure. While a small cluster of points still falls below the linear trend, we suspect this is due to some error in estimating the vectors that define the rhomboidal flat torus.

Finally, note that in all three plots we have a positive intercept. As discussed in Section 5, assuming our random function model for the data, this offset is due to the positive noise term $\sigma$ and its appearance in equation (7), recalled here:

$$\frac{1}{p}\|\mathbf{Y}_i - \mathbf{Y}_j\|^2 \approx \frac{1}{p}\|\phi(z_i) - \phi(z_j)\|^2 + 2\sigma^2.$$

The moving averages, plotted in red, have been computed using a window size of $0.01n = 150$, with the shaded region showing $\pm 1$ standard deviation. The results are not sensitive to the value of $k$ used, with the correlation coefficient staying within $[0.73, 0.75]$ for Model 3 for reasonable values of $k \in [6, 20]$.

## H.3. Alternative model

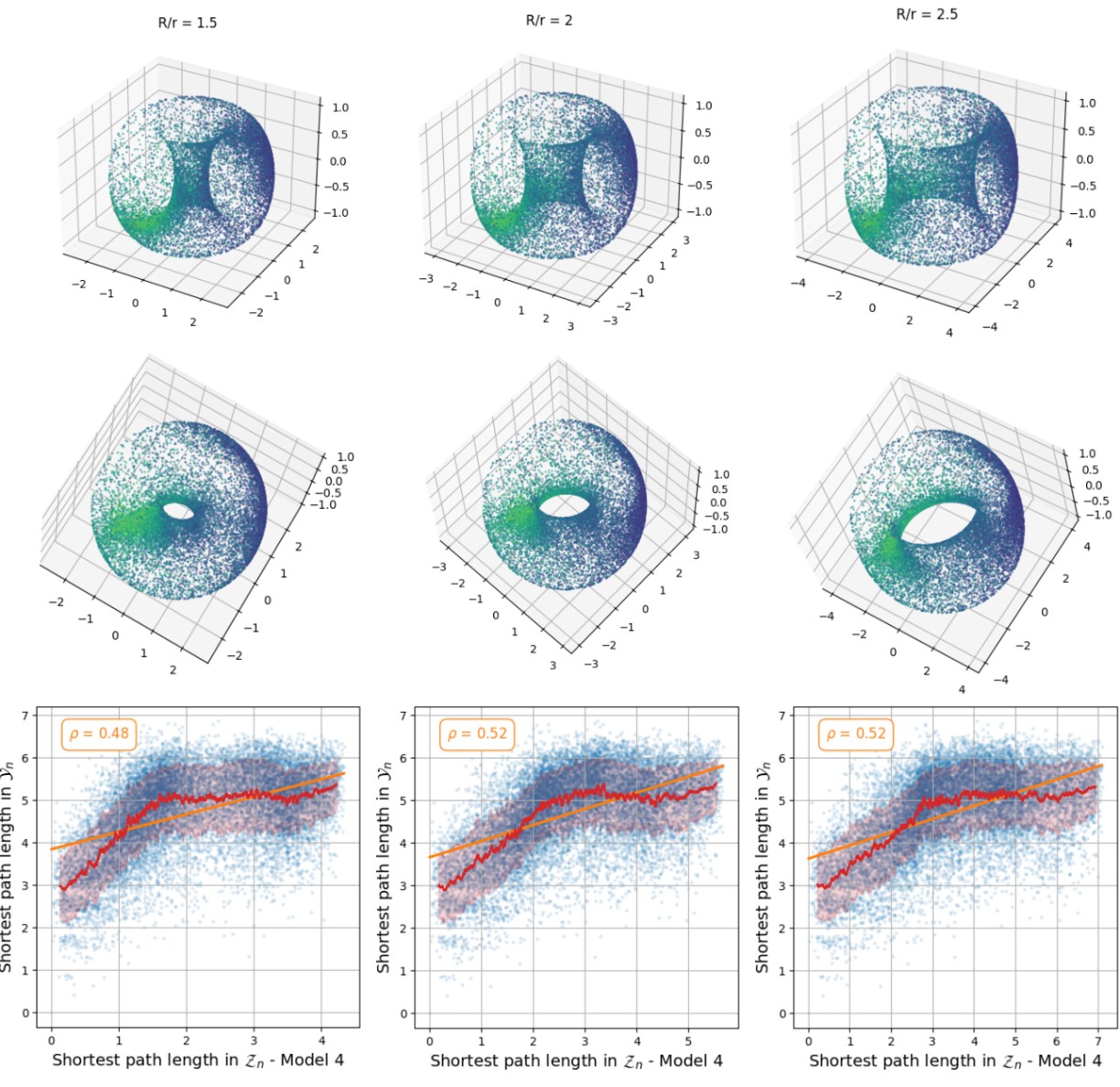

*Figure 8.* The top two rows show $z_i$ under Model 4 for three different ratios of torus radii $R/r \in \{1.5, 2, 2.5\}$ (left to right). The bottom row shows distance-distance plots comparing the shortest paths on $\mathcal{Y}_n$ and $\mathcal{Z}_n$ for each torus. Unlike Model 3, these embeddings exhibit a pronounced deviation from linearity, especially at larger distances.

Building on the three models discussed in Section H.2, we introduce a fourth: in 'Model 4', the $z_i$ are points on a torus embedded in $\mathbb{R}^3$. We construct this model by taking the decoded toroidal coordinates from Gardner et al. (2022) and mapping them onto a standard 3D torus. The geometry of a 3D torus is not only governed by its circular coordinates, but also by the ratio of the two radii: the larger radius $R$, which defines the circular path around around the central cavity, and the smaller radius $r$, which defines the cross-section of the tube.

The top two rows of Figure 8 show the $z_i$ under Model 4 for three different values of the ratio $R/r \in \{1.5, 2, 2.5\}$. The bottom row plots the shortest path lengths on $\mathcal{Y}_n$ against the shortest path lengths on $\mathcal{Z}_n$, which were calculated in the same way as for Models 1-3. We see that the linear fit under Model 4 is much worse than Model 3 in the main text.

The fact that there exists no $C^2$ (twice continuously differentiable) isometric embedding of a flat torus into $\mathbb{R}^3$ (proved by (Hartman & Nirenberg, 1959), for example) means that shortest paths (and hence shortest path-lengths) in $\mathcal{Z}$ under Model 3 are not equal to those under Model 4. Assuming that $\mathcal{Z}$ from Model 3, i.e. the rhomboidal flat torus, is indeed isometric to $\mathcal{M}$, the non-isometry between $\mathcal{Z}$ under Models 3 and 4 explains the non-isometric relationship between distances on $\mathcal{M}$ and $\mathcal{Z}$ from Model 4, observed in the bottom row of Figure 8. Intuitively, while distances may be preserved on the initial "roll up" of the rhombus into a tube, the final step — joining the tube's ends to form a torus — must involve stretching/compressing, depending on the radii of the torus. Suppose that the tube is only stretched (and not compressed) to form the torus; this distortion would inflate distances on $\mathcal{Z}$ from Model 4 (compared to Model 3), providing an explanation for the observed deviation from linearity in the bottom row plots of Figure 8.

# I. Estimating $p_{\text{eff}}$

Here we investigate the effect of finite sample size on estimation of $p_{\text{eff}}$. We simulate independent samples from a multivariate Gaussian with $p = 100$. We define the covariance matrix such that the first 20 eigenvalues are 1 and the remaining 80 are 0.1, resulting in an effective dimension $p_{\text{eff}} = 28$. As shown in Figure 9, the estimator $\widehat{p}_{\text{eff}}$ is systematically biased downwards, but converges to the true value as $n \to \infty$. This shows that the estimates reported in Table 1 serve as conservative lower bounds, which further supports our claim that the dimensionality of these datasets is sufficient to induce concentration about the underlying latent structure.

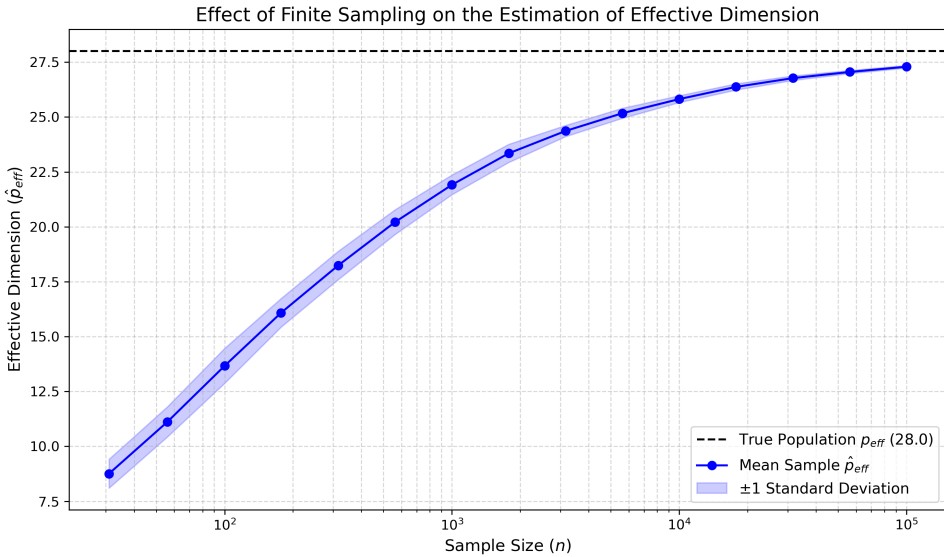

*Figure 9.* We simulate $n$ independent samples from a multivariate Gaussian distribution ($p = 100$) with a known effective dimension of $p_{\text{eff}} = 28$ (black dashed line). The sample estimator $\hat{p}_{\text{eff}}$ (blue line) exhibits a systematic downward bias at small sample sizes, asymptotically converging to the true value as $n$ grows. Shaded regions denote $\pm 1$ standard deviation over 50 independent trials.

## J. Dataset descriptions

Here we provide brief descriptions of the datasets used in our empirical estimation of $p_{\text{eff}}$, covering a range of data modalities and application domains.

**Planaria**: a single-cell gene expression dataset from adult planarians (a type of flatworm) (Plass et al., 2018), where each observation corresponds to an individual cell represented by gene-level expression features.

**Newsgroups**: a text dataset consisting of documents from the 20 Newsgroups corpus (Lang, 1995), represented using TF-IDF features.

**Amazon reviews**: a text dataset consisting of reviews of Amazon products, represented using TF-IDF features (Kashnitsky, 2020).

**MNIST**: a dataset of handwritten digit images (of the digits 0 -9) (LeCun et al., 2002), where each image is represented as a vector of grayscale pixel intensities.

**S&P 500**: a multivariate financial time series consisting of daily returns for stocks in the S&P 500 index over a five year window (Cam Nugent, 2020).

## Appendix References

Cam Nugent. S&p 500. https://www.kaggle.com/datasets/camnugent/sandp500, 2020. Accessed: 2025-01-26.

Carter, K. M., Raich, R., and Hero III, A. O. On local intrinsic dimension estimation and its applications. *IEEE Transactions on Signal Processing*, 58(2):650–663, 2009.

Cayton, L. Algorithms for manifold learning. *Univ. of California at San Diego Tech. Rep*, 12(1-17):1, 2005.

De Silva, V. and Vejdemo-Johansson, M. Persistent cohomology and circular coordinates. In *Proceedings of the twenty-fifth annual symposium on Computational geometry*, pp. 227–236, 2009.

Dijkstra, E. W. A note on two problems in connexion with graphs. *Numerische mathematik*, 1(1):269–271, 1959.

Donoho, D. L. High-dimensional data analysis: The curses and blessings of dimensionality. *Invited lecture at Mathematical Challenges of the 21st Century*, 2000. AMS National Meeting, Los Angeles, CA, USA, August 6-12,.

Hartman, P. and Nirenberg, L. On spherical image maps whose jacobians do not change sign. *American Journal of Mathematics*, 81(4):901–920, 1959.

Hein, M. and Audibert, J.-Y. Intrinsic dimensionality estimation of submanifolds in rd. In *Proceedings of the 22nd international conference on Machine learning*, pp. 289–296, 2005.

Kainen[1], P. C. Utilizing geometric anomalies of high dimension: When complexity makes computation easier. *Computer Intensive Methods in Control and Signal Processing: The Curse of Dimensionality*, pp. 283, 1997.

Kashnitsky, Y. Hierarchical text classification dataset. https://www.kaggle.com/datasets/kashnitsky/hierarchical-text-classification, 2020. URL https://www.kaggle.com/datasets/kashnitsky/hierarchical-text-classification. Accessed: 2026-01-27.

Kégl, B. Intrinsic dimension estimation using packing numbers. *Advances in neural information processing systems*, 15, 2002.

Lang, K. Newsweeder: Learning to filter netnews. In *Machine learning proceedings 1995*, pp. 331–339. Elsevier, 1995.

Lawrence, N. Spectral dimensionality reduction via maximum entropy. In *Proceedings of the Fourteenth International Conference on Artificial Intelligence and Statistics*, pp. 51–59. JMLR Workshop and Conference Proceedings, 2011.

LeCun, Y., Bottou, L., Bengio, Y., and Haffner, P. Gradient-based learning applied to document recognition. *Proceedings of the IEEE*, 86(11):2278–2324, 2002.

Levina, E. and Bickel, P. Maximum likelihood estimation of intrinsic dimension. *Advances in neural information processing systems*, 17, 2004.

Little, A. V. *Estimating the intrinsic dimension of high-dimensional data sets: a multiscale, geometric approach*. PhD thesis, Duke University, 2011.

Plass, M., Solana, J., Wolf, F. A., Ayoub, S., Misios, A., Glažar, P., Obermayer, B., Theis, F. J., Kocks, C., and Rajewsky, N. Cell type atlas and lineage tree of a whole complex animal by single-cell transcriptomics. *Science*, 360(6391):eaaq1723, 2018.

Sutherland, W. A. *Introduction to metric and topological spaces*. Oxford University Press, 2009.

Tibshirani, R. Regression shrinkage and selection via the lasso. *Journal of the Royal Statistical Society Series B: Statistical Methodology*, 58(1):267–288, 1996.

