# OpenReview forum: "How High is ‘High’? Rethinking the Roles of Dimensionality in Topological Data Analysis and Manifold Learning"
_ICML.cc/2026/Conference — ICML 2026 regular_

### Official Review · Reviewer_ND9E · 2026-03-11

**Soundness:** 2
**Presentation:** 3
**Significance:** 3
**Originality:** 3
**Overall Recommendation:** 4
**Confidence:** 3

**Summary:**

This paper studies the geometry of high-dimensional data and argues that high dimensionality can sometimes be a statistical advantage rather than an obstacle. The authors introduce a random function model and define the concept of effective dimension $p_{eff}$. They prove that if the effective dimension grows faster than $\log n$, i.e. $p_{eff} \in \omega(\log n)$, the dot products of the data points will concentrate around their expected values. The paper then shows that under these conditions, Topological Data Analysis (TDA) methods can successfully recover the latent manifold structure of the data. Finally, the authors apply this theory to a neuroscience dataset i.e., grid cell activity, to find evidence of a toroidal structure.

**Compliance With Llm Reviewing Policy:**

Affirmed.

**Final Justification:**

The author rebuttal successfully addressed my technical and practical concerns, specifically regarding error bounds on manifold distances and the validity of sub-Gaussian assumptions in biological data. And I will maintain my original positive score.

**Key Questions For Authors:**

1. In Prop 3.2, the authors prove the concentration of pairwise Euclidean distances around their expected values. However, in Section 5, hte empirical evidence for global isometry relies on computing geodesic distances via shortest paths on knn graph. Since Dijkstra's algotihm sums up multiple local Euclidean distances along a path, the concentration errors could potentially accumulate. Is there a guarantee or bound in your framework that proves the cumulative error along the path remains bounded?

2. Theorems 2.1 and 3.1 depend on Assumptions A1 and A2 (zero-mean and sub-Gaussian distributions). How do you justify applying this specific theory to your experimental data?

**Limitations:**

yes

**Strengths And Weaknesses:**

Strengths:
1. The theoretical definitions are clean and make sense. Separating the raw feature dimension $p$ from the effective dimension $p_{eff}$ provides a better way to measure the true variability of the data.
2. The paper connects different areas well. Combining statistical inequalities (generalized Hanson-Wright) with topological data analysis and testing it on real neuroscience data is an interesting approach.

Weaknesses:
1. The main theorems (Theorem 2.1 and 3.1) strongly rely on Assumptions A1 and A2, which require the data and noise to be zero-mean and sub-Gaussian. However, the main experiment uses grid cell firing rates. The authors do not explain why it is okay to apply a zero-mean, sub-Gaussian theory to this type of biological data.

2. In Section 5, the authors use a k-nearest neighbor graph to approximate shortest paths on the manifold and report a linear regression correlation of $\rho = 0.75$. The paper does not discuss how the choice of k affects this result, making it hard to know if the finding is stable.

---

> ### Author Rebuttal · Authors · 2026-03-30
>
> Thank you for your review. We are glad that you found the interdisciplinary connections interesting and the theory clear. We address your questions below.
>
> > **Q1**  *``Since Dijkstra's algorithm sums up multiple local Euclidean distances... is there a guarantee or bound in your framework that proves the cumulative error along the path remains bounded?''*
>
> Using Theorem 3.1, let the maximum distance error on any single edge in the $k$-NN graph be bounded by $\varepsilon = O_\mathbb{P}\left(\sqrt{\log n / \min_{i \in [n]}p_\mathrm{eff}^{(i)}} \right)$. Then in the calculation of a path containing $L$ hops, the maximum cumulative error is bounded by $L \varepsilon$. Since the shortest paths on these manifolds require $L \ll n$ hops, the cumulative error remains well-controlled.
>
> > **Q2** *``Theorems 2.1 and 3.1 depend on Assumptions A1 and A2 (zero-mean and sub-Gaussian distributions). How do you justify applying this specific theory to your experimental data?''*
>
> In biological neural networks, neurons have physiological upper limits on their maximum firing rates due to refractory periods (the time after a neuron fires during which it cannot fire again), and mathematically, any strictly bounded random variable is inherently sub-Gaussian.
> Regarding assumption **A2** (zero mean), this applies specifically to the noise fluctuations ($\mathbf{E}_i$). If there are noise biases, e.g. due to measurement tools consistently under-measuring, that bias would be absorbed by the mean vector.
>
> > *``The paper does not discuss how the choice of $k$ affects this result, making it hard to know if the finding is stable.''*
>
> We appreciate that demonstrating robustness to hyperparameters is important. The minimum $k$ to ensure a connected graph is $k=6$, which gives $\rho=0.74$. In our experiment we used $k=10$, leading to $\rho=0.75$. Doubling this to $k=20$, gives $\rho=0.73$, demonstrating that reasonable values of $k$ in the range [6,20] lead to stable estimates of $\rho$. We will add this detail in the Appendix.

---

> > ### Author Rebuttal · Reviewer_ND9E · 2026-04-02
> >
> > Thank you for the clear rebuttal. It directly addresses my questions.
> > ​
> > First, the L_epsilon bound for the cumulative error is reasonable. Since the shortest path length L \ll n, the error accumulation on the manifold is effectively controlled.
> >
> > ​Second, using physiological limits to justify the sub-Gaussian assumption makes practical sense. Clarifying that the zero-mean assumption strictly applies to the noise term E_i resolves my concerns about applying the theory to real biological data.
> > ​
> > Third, the new empirical results show that the correlation is stable \rho \in [0.73, 0.75] for reasonable values of k (6 to 20). This confirms the finding is not highly sensitive to the choice of k.
> >
> > ​Thank you for the clear rebuttal, which addresses my questions and I will maintain my score.

---

### Official Review · Reviewer_8GFm · 2026-03-11

**Soundness:** 2
**Presentation:** 1
**Significance:** 3
**Originality:** 3
**Overall Recommendation:** 4
**Confidence:** 3

**Summary:**

The paper analyses a beneficial effect of high-dimensionality on topological data analysis and manifold learning. It extends the Hanson-Wright inequality to a specific model of high-dimensional data with possible dependence structure and thus establishes concentration bounds that grow like the logarithm of the dataset size over the effective dimension of the dataset. Using this, they explain why persistent homology recovers latent homology in regimes where the effective dimension dominates the the logarithm of the dataset size. Finally, they revisit the grid cell data of Gardner et al. and show that its toroidal structure is in fact close to isometric to a flat torus.

**Compliance With Llm Reviewing Policy:**

Affirmed.

**Final Justification:**

The rebuttal helped to resolve a major misunderstanding on my side and I now appreciate the central point of the paper. Some small aspects are still unclear, e.g., the exact effect of the noise on birth and death times as well as the interaction of PCA and the proposed theory when applied to the Gardner et al. data. As a result, I increased my score to 4 and my confidence to 3.

**Key Questions For Authors:**

Q1 Eq (7): Even under the assumptions before Eq (7) I do not understand how its statement can be derived from Thm 3.2. The latter is about dot products and makes no statements about norms, which are needed to get to squared distances. Moreover, I do not understand if equation 7 deem good or bad. It seems the authors want to claim that the true data structure gets recovered, but the constant term $2\sigma^2$ completely dominates for high $p$. This is precisely the problematic nature of high-dimensionality.

Q2 Blessing for PCA: Given the interesting toy experiment in Sec 6.1, where (something like) PCA becomes better the more noise dimensions get added, I wonder if one can improve any PCA by artificially adding noise dimensions? Empirically, I find PCA to perform worse in the presence of many noise dimensions.

**Limitations:**

The sub-Gaussian assumption is discussed, but assuming my points on soundness are valid, I think a summary of them should be added as limitations.

**Strengths And Weaknesses:**

**Soundness**
I did not check the proofs in the appendix.

*Pro:*
The linear regression test for isometry in Sec 6.3 is convincing.


*Con:*
Usefulness of asymptotic statements. The theorems in Section 3 and 4 are asymptotic in nature and only recover the latent structure if $\log(n)/p_{eff} \to 0$. The authors show in Sec 6.2 that for many real-world datasets $\log(n)$ and $p_{eff}$ are roughly similar, but I do not see why that should imply decent concentration on any concrete finite dataset given that the $O_P$ notation hides a potentially very large constant $M$.

Consistency of persistence diagrams: The authors mention the constant error term $\sigma^2$ in last inequality of Sec 4, but still claim that PH finds the correct structure. Why can one ignore the error term $\sigma^2$ which does not even vanish asymptotically?


PH computation for grid cells. To compute the persistent homology of the grid cell data, the authors follow Gardner et al. and massively reduce the dimensionality from 150 to 6. This seems to go against the theoretical point that higher dimensions can help.

The toy example in Sec 6 uses Gaussian errors $E$, while the theory assumes sub-Gaussian errors. Why this inconsistency? It would also be nice to compute the errors between noisy and true structure as in Thm 3.2 on that toy data and compare it against the prediction of Thm 3.2.

**Presentation**
*Pro:*
The toy example in section 6 is very helpful to illustrate the main theoretical points. I think it would be nice to flesh it out as mentioned above.

*Con:*
Section 3 and 4 are very dense and do not contain sufficient motivation. The high-dimension low sample size situation is typically viewed as problematic, so statement like "$p/n \to \infty$ is not necessary" (line 077 left) are confusing. The usual issue is that any data concentrates around a default layout (equal distances, equal norms, everything orthogonal), which does not encode its true, latent structure. It would therefore be very helpful to provide intuition on why higher dimensionality and the ensuing concentration phenomenon should be beneficial. In a similar vein, it would be helpful to discuss how difficult it is to create data generative processes following the proposed model such that $p_{eff} \in \omega\log(n))$. The toy example is one instance, but a more general treatment would help to appreciate the applicability of the analysis.

Disconnect between theory section and grid cell section. How is the theory from section 3 and 4 applied to grid cells? Is the functional form in line 360 right fit or verified for the grid cell data?

Missing references: Recent work described a curse of dimensionality for persistent homology, both theoretically and empirically. This is especially relevant since section 4 seems to claim that the curse of dimensionality is not actually pronounced for persistent homology.
Hiraoka et al. Curse of Dimensionality on Persistence Diagrams 2024
Damrich et al. Persistent Homology for High-dimensional Data Based on Spectral Methods, 2023



**Significance**
*Pro:*
The provided explanation for why the curse of dimensionality might not affect practical applications as much as one might think provides a useful perspective on high-dimensional data that is potentially of interest of the wider ML community.

The isometry results for the grid cells are a very interesting extension of Gardner et al.'s analysis.

*Con:*
It would be helpful to discuss how common the proposed model (line 156 left) is for real data.


**Originality**
*Pro:*
The proposed model of high-dimensional data in line 156 left is an interesting generalization of the setup of Whitely et al.

---

> ### Author Rebuttal · Authors · 2026-03-30
>
> Thank you for your review and highlighting the usefulness of our geometric perspective for the wider ML community. We appreciate your confidence score and believe your main concerns stem from some common misconceptions, which we address below.
>
> > **Q1** *Even under the assumptions before Eq (7) I do not understand how its statement can be derived from Thm 3.2.  The latter is about dot products and makes no statements about norms, which are needed to get to squared distances... the constant term $2\sigma^2$ completely dominates for high $p$... [in PH] Why can one ignore the error term $\sigma^2$ which does not even vanish... It would be helpful to provide intuition...*
>
> Regarding the derivation of (7), for any $i \neq j$, the squared distance is $||Y_i - Y_j||^2 = ||Y_i||^2 + ||Y_j||^2 - 2 \langle Y_i, Y_j \rangle$. As established in Eq. (6) of the main text, the expected dot product for any $i,j$ is $\mathbb{E}[\mathbf{Y}_i \cdot \mathbf{Y}_j] = \phi(z_i) \cdot \phi(z_j) + p \sigma^2 \mathbf{I}[i=j]$. Prop 3.2 uses this expectation to show that the cross-term $\frac{1}{p}\langle Y_i, Y_j \rangle$ concentrates around the latent signal $\frac{1}{p}\langle \phi(z_i), \phi(z_j) \rangle$, and that $\frac{1}{p}||Y_i||^2 = \frac{1}{p}\langle Y_i, Y_i \rangle$ concentrates around $\frac{1}{p}||\phi(z_i)||^2 + \sigma^2$. Substituting these back into the squared distance expansion gives Eq. (7).
>
> To provide intuition on why high dimensionality is beneficial, one can view the scaled dot product as an empirical average where a law of large numbers argument applies, evaluated across the $p$ dimensions. For distinct points ($i \neq j$), as $p$ grows, the independent noise components average out to zero, leaving behind the latent inner product structure. Conversely, for the norm of each point ($i = j$), the noise variance averages to $\sigma^2$. Geometrically, high-dimensional noise vectors are mutually orthogonal; they deterministically inflate magnitudes by $\sigma^2$ rather than scrambling the points, effectively 'pushing' the latent manifold onto a high-dimensional hypersphere.
>
> Regarding persistent homology, the $2\sigma^2$ term is a global, uniform shift added to all squared distances. It perfectly preserves distance ordering (nearest neighbors) and simply delays PH birth/death times by a constant, leaving the latent topological structure intact.
>
> **Q2: Adding noise dimensions**
>
> Firstly, the toy experiment in Sec 6.1 does not add pure noise dimensions. We increase the ambient dimension $p$ (and $p_\mathrm{eff}$) by observing more features from the generative model. Each new dimension contains both a signal and a noise component.
>
> You are absolutely right that adding pure noise dimensions ruins PCA. However, in our model, assumption **A3** (mean-square continuity) ensures that as $p$ grows, the added dimensions are *informative* (not pure noise).
> In the current manuscript we say "Assumption **A3**  distinguishes $\mathbf{Y}^{\mathrm{nf}}(z_i)$ as "signal'' rather than noise, cf. $\mathbf{E}_i$ being independent across $i$ under **A2**.'' In the revision we will clarify that **A3** "distinguishes *each dimension* of $\mathbf{Y}^{\mathrm{nf}}(z_i)$ as providing "signal'' rather than pure noise, ensuring the informative signal grows with $p$''. Moreover, we already state in the abstract "under mild continuity assumptions (ensuring that features bring additional information as dimension grows)''.
>
> **Asymptotic vs. finite sample statements.**
> The results in Section 3 and 4 are not purely asymptotic; they are finite-sample concentration inequalities. As shown in our new Appendix simulation (provided in response to Q1 Reviewer BVaR) the scaling constant behind the $O_\mathbb{P}$ notation is empirically small, and our concentration bound accurately reflects the empirical error.
>
> **Dimensionality reduction**
> PCA is used to reduce dimensionality because the Ripser (PH) algorithm computationally struggles with high-dimensions. For the experiment in Figure 4 (in search of isometry), we computed shortest paths directly on the raw, 149-dimensional data.
>
> **Gaussian errors**
> There is no inconsistency here. A sub-Gaussian random variable is defined as having tails that decay at least as fast as a Gaussian distribution.
>
> **Plausibility of** $p_{\text{eff}} \in \omega(\log n)$ This is  not pathological. It simply requires new features to span diverse, orthogonal functional directions.
>
> **Grid cell disconnect.**
> The model (line 360) directly reflects established neuroscience: grid cells fire in repeating hexagonal lattices (topologically a torus) corresponding to the animals physical location.
>
> **Missing references.**
> Thank you for pointing out Hiraoka et al. (2024) and Damrich et al. (2023). We will cite and discuss them in the revision. They model the "curse" where high dimensions are mostly noise. We complement them by showing the "blessing" when high dimensions provide informative features.

---

> > ### Author Rebuttal · Reviewer_8GFm · 2026-04-02
> >
> > Many thanks for the very helpful rebuttal. Especially the emphasis on the fact that new dimensions are assumed to include additional signal was very helpful! This also resolves the apparent contradiction between the present paper and the two references, I provided (though they still seem related). I have some small remaining questions:
> >
> > **Dimensionality reduction**
> > I do not understand the concern here. Why should ripser struggle with high-dimensionality? It simply computes the pairwise distance matrix, which scales linearly in the dimension. Alternatively, one could just precompute the distance matrix and give it to ripser, to completely hide the original dimensionality.
> >
> > **Plausibility of $p_{eff} \in w(\log n)$**
> > I did not understand your explanation, which did not mention the dataset size at all. The formula clearly seems to depend on $n$.
> >
> > **Grid cell disconnect**
> > I appreciate that grid cells are known to approximately fire in a toroidal pattern. But why this approximation is precisely captured by the formula in line 360 and also satisfies assumptions A1-A4, I do not see.
> >
> > **Persistent homology**
> > The statement about birth and death times being shifted by exactly $\sigma^2$ seems like a stronger statement than the statement about bottleneck distance. If the authors believe their arguments prove this stronger statement, then I would recommend phrasing the result also in the stronger form and detailing the proof.

---

> > > ### Author Response · Authors · 2026-04-07
> > >
> > > Thanks for the thought-provoking comments and the opportunity to clarify further.
> > >
> > > **Dimension reduction** I apologise for causing confusion here, I made a mistake in my response above --- it is the number of samples $n$ that ripser struggles with computationally, rather than $p$. The primary reason we apply PCA in this case is to recreate the Gardner et al. analysis. For contrast, here is an anonymous link to TDA on the raw, high-dimensional data (https://ibb.co/XhVW8Vd). It identifies the correct $H_0$ and $H_1$, however the noise seems to close the 2D cavity early. In this sense, a PCA step seems useful for TDA.
> > >
> > > We think this doesn't contradict the main messages of our work; in future we hope to conduct more detailed theoretical analysis of PCA under our model, but just as a rough pointer for now, we note that application of PCA for dimension reduction of a $n\times p$ data matrix $\mathbf{Y}\equiv[\mathbf{Y}_1|\ldots|\mathbf{Y}_n]^{\top}$ (from $p$ dimensions to $s\leq p$ dimensions) outputs the rows of the $n\times s$ matrix $\mathbf{U}_s\boldsymbol{\Lambda}_s^{1/2}$  where columns of $\mathbf{U}_s$ are the top $s$ orthonormal eigenvectors of $\mathbf{Y}\mathbf{Y}^{\top}$ and $\boldsymbol{\Lambda}_s$ is the diagonal matrix of associated eigenvalues. The crucial point here is that the $(i,j)$th element of $\mathbf{Y}\mathbf{Y}^{\top}$ is the dot-product $\mathbf{Y}_i\cdot\mathbf{Y}_j$, to which our theory applies. We conjecture that PCA is able to `remove' the additive noise in our model which impacts the diagonal of $\mathbb{E}[\mathbf{Y}\mathbf{Y}^{\top}]$, but further analysis is beyond the scope of the present work.
> > >
> > > To illustrate failure of a much less `smart' approach to dimension reduction, here is a link to TDA on just the first six dimensions of the data (https://ibb.co/mjLdkJ9) --- there is no visible topological structure at all.
> > >
> > > Incidentally, we note our isometry analysis in Figure 4 is computed directly on the raw, high-dimensional data. There is still a considerable amount of noise around the linear fit,  perhaps due to moderate (rather than large) estimated $p_\mathrm{eff}$.
> > >
> > > **Plausibility of $p_\mathrm{eff} \in \omega(\log n)$**
> > > We apologise for a very short explanation earlier, we ran out of characters!
> > >
> > > First, let us give a simple illustration of the regime $p_\mathrm{eff} = p \in \omega(\log n)$.
> > > Starting from l. 142, consider $n$ to be growing and $p$ to be any sequence growing with $n$ such that $p \in  \omega(\log n)$. Then let each $X_j$, $j=1,2,\ldots,$ be (say) i.i.d. random functions over $Z$ (e.g., Gaussian processes satisfying A1) and set $\Sigma = I$, then $p_\mathrm{eff} = p$  as required. We note that that $\Sigma$ implicitly depends on $p$ and hence $n$.
> > >
> > > Now, for a simple example which satisfies $p_\mathrm{eff}<p$ and at the same time $p_\mathrm{eff} \in \Theta(p) \in \omega(\log n)$, we can construct $p_\mathrm{eff} = \alpha p$ for some fixed $\alpha \in (0,1)$, as follows.
> > > With $n$ growing and $p \in \omega( \log n)$ as before, set $\mathbf{\Sigma} = \mathrm{diag}(\lambda_1, \ldots, \lambda_p)$ where $\lambda_1 = 1$; and for all $p$ large enough that $\alpha p> 1$ (recall $\alpha\in(0,1)$ is fixed), let $\lambda_2,\ldots,\lambda_p$ be such that $\sum_{i=2}^p \lambda_i = \alpha p -1$, e.g., $\lambda_i=(\alpha p-1)/(p-1)$. Then $p_\mathrm{eff} = 1 + \alpha p - 1 = \alpha p$, as required.
> > >
> > >
> > > **Grid cell disconnect.** Informed by the analysis of Gardner et al, our model is intended to simply capture the idea that firing data can be explained in terms of a latent state $z_i$ on the latent torus, mapped to high-dimensional neural firing rates $Y_i$. We explain verbally in the paper what each of the elements in the model correspond to in this specific example, just after line 360 (2nd column). We will add a sentence acknowledging that this model (like all models) is, of course, an abstraction of reality. Regarding our theoretical assumptions (A1-A4): see the response to reviewer ND9E (Q2) for A1-A2. For A3, we believe it is reasonable to assume that the expected firing rate $\mathbf{\mu}(z)$ and the correlated modulations $\mathbf{\Sigma}(z)^{1/2}\mathbf{X}(z)$ vary continuously with changes in physical (and thus toroidal) space because we are not aware of any biological knowledge or existing empirical results suggesting they should be discontinuous. For A4, the latent space is a torus, which is compact.
> > >
> > > **Persistent homology** I apologise for the confusion here. To clarify, the $2\sigma^2$ shift applies to the \textit{squared} distances. For the standard distances (used in PH) this results in a non-linear but strictly monotonic transformation (, distances between data vectors behave as $d_\text{obs} \approx \sqrt{d_\text{latent}^2 + 2\sigma^2}$), and thus the ordering of neighbours, the sequence of birth and death times, and therefore the general topological structure is preserved. The actual birth/death times are warped non-linearly rather than simply shifted.

---

### Official Review · Reviewer_KaQK · 2026-03-11

**Soundness:** 3
**Presentation:** 3
**Significance:** 4
**Originality:** 4
**Overall Recommendation:** 5
**Confidence:** 3

**Summary:**

The present work adapts several inequalities to the study of the geometry and topology of manifolds realized through high-dimensional random functions. Several of the results are notable. First off, if the effective dimensionality, defined in terms of the normalized trace of the covarince matrix, is asymptotically bounded by $\log n$, then random vectors tend to be orthogonal to each other. Secondly, under a few assumptions the authors introduce a random function model with which they can determine if the number of features and samples are sufficient to recover the homology groups of the latent dynamics of a system. Finally, the authors study grid cell topology and geometry arguing that the transformation from space to the grid cell torus is an isometry.

**Compliance With Llm Reviewing Policy:**

Affirmed.

**Final Justification:**

The rebuttal addressed my questions in a satisfying manner. I therefore will stick with my positive score.

**Key Questions For Authors:**

*Major comments*
1. You make assumption 5 (the same applies to assumption 6 in the appendix) to establish a homeomorphism between $\mathcal{Z}$ and $\mathcal{M}$. However, many interesting transformations in machine learning are not injective and that is exactly what makes them interesting. For example every classification task is like this. Another example is the map from physical space to the grid cell torus (I understand that under your third model of space they are homeomorphic). Can you say anything about the relation between diag$(\mathcal{Z})$ and diag$(\mathcal{M})$ when this assumption does not apply?

2. From Figure 4c you conclude that there is an isometry between space and the torus and you say that "The deviation from linearity at larger distances is likely due to estimation error in cohomological decoding.." (paraphrased). Can you elaborate on why this is the reason further? My intuition is that this result acutally points in the opposite direction. Namely it is evidence that the isometry is only local and globally the map is not an isometry.

3. When comparing the shortest paths on the torus to your models, do you use the preprocessed version of the grid cell data (PCA+UMAP) or is that just for the plots of the torus? If so, shouldn't an effective dimensionality of 12.67 and $\log n = 9.62$ be enough to not require the PCA+UMAP dimensionality reduction?

*Minor comments*
- On the second column of line 210, should $p$ always be the same for $\mathcal{Y}$ and $\mathcal{M}$?
- In table 1, for MNIST you have n=5000, but MNIST has a total of 70000 images. You say that the estimates include *all* samples, so is this a typo?

**Limitations:**

Yes

**Strengths And Weaknesses:**

**Strengths**
- The topic is very interesting. Understanding the topology and geometry of high-dimensional noisy neural manifolds is essential to furhter neuroscience as a field.
- The paper is well written and the proofs in the Appendix provide justification for all the claims in the paper.
- The findings of the paper are likely going to be significant contributions.

**Weaknesses**
- A few assumptions, namely assumption 5 is a bit strong and many interesting problems in machine learning do not obey it.
- The conclusions for isometry of the grid cell torus are questionable.

---

> ### Author Rebuttal · Authors · 2026-03-30
>
> Thank you for the strong support of our work and for recognising the significance of our findings. We are glad that you found the paper well-written and the topic interesting. We address your questions in detail below.
>
> > **1. Assumption 5 and non-injective maps.** *"Can you say anything about the relation between $dgm(\mathcal{Z})$ and $dgm(\mathcal{M})$ when this assumption does not apply?''*
>
> When **A5** is relaxed, $\phi$ is no longer guaranteed to be a homeomorphism, and $\mathrm{dgm}(\mathcal{Z})$ and $\mathrm{dgm}(\mathcal{M})$ will no longer necessarily match. Your classification example illustrates this well. A perfect classification mapping would collapse continuous regions of $\mathcal{Z}$ into discrete points (the classes), in which case $\mathrm{dgm}(\mathcal{M})$ will simply give $H_0$ equal to the number of classes, destroying any higher order homology of $\mathcal{Z}$.
>
> > **2. Global vs local isometry (Figure 4c).** *"Can you elaborate on why [cohomological decoding error] is the reason further?''*
>
> We appreciate the reviewers intuition that a deviation at larger distances often indicates a transition from global to local isometry. However, in this specific case, we believe that the downward deviation at the extreme right is a measurement artifact arising from the error in the decoded vectors, $r_1$ and $r_2$, defining the flat torus rhombus. We suspect that the observed 'droop' indicates that the decoded rhombus is slightly over-estimated in length (along the long cross-diagonal). Thus, distances to the far corners are over-estimated in $\mathcal{Z}_n$, relative to the neural manifold $\mathcal{Y}_n$. Moreover, as shown in Appendix G.2, under Models 1 and 2, where the relationship is locally isometric (but not globally isometric), the deviation from linear after the distance where isometry fails is significantly more pronounced. In Model 3, the fact that linearity is tightly maintained until the very boundaries of the rhombus strongly supports our conclusion of global isometry.
>
> **3. Preprocessing (PCA + UMAP).**
> We did not use PCA + UMAP to compute the shortest paths. As detailed in Apendix G.1 and G.2, PCA and UMAP were only used for the 3D visual plotting (Figure 3a) and to reduce the computational burden for the persistent homology algorithm (Figure 3c). We apologise that this was buried in the Appendix and will add in the main text of Section 6.3 that the geodesic path calculations were performed directly on the ambient data.
>
> **Minor comments:**
> - Line 210. Yes, $p$ is the same for both $\mathcal{Y}_n$ and $\mathcal{M}$. The manifold $\mathcal{M}$ is defined as the image of $\phi: \mathcal{Z} \to \mathbb{R}^p$, and each observation $\mathbf{Y}_i \in \mathbb{R}^p$.
> - MNIST $n=5000$. It is true that we used a random 5000-sample subset to compute the sample covariance efficiently. What we meant by that statement is that the estimator is based on the sample covariance, which is calculated over $n$ samples. We will make this clearer.

---

> > ### Author Rebuttal · Reviewer_KaQK · 2026-04-01
> >
> > Thank you for your response. I enjoyed the paper and believe it should be accepted. I will maintain my positive score.

---

### Official Review · Reviewer_BVaR · 2026-03-18

**Soundness:** 3
**Presentation:** 3
**Significance:** 3
**Originality:** 3
**Overall Recommendation:** 5
**Confidence:** 3

**Summary:**

This submission develops a statistical framework that revisits the role of dimensionality in manifold learning and topological data analysis (TDA). The authors (i) introduce a generalized Hanson–Wright (GHW) concentration inequality for possibly dependent sub-Gaussian feature vectors, (ii) distinguish three notions of dimension—effective dimension p_eff, correlation rank r, and intrinsic dimension d—and (iii) show that, when p_eff >> log n, pairwise inner products and distances concentrate around their expectations, thereby revealing the underlying manifold and topological structure.

They further establish consistency results for persistent homology in this regime and support their theory with simulations and a reanalysis of grid-cell neural data, suggesting that neural population activity is approximately isometric to a flat torus model of physical space.

**Compliance With Llm Reviewing Policy:**

Affirmed.

**Final Justification:**

The authors properly adressed my concerns. I think this new view about the role of dimensionality has both the technical level and the importance to be published in ICML

**Key Questions For Authors:**

1) What happens out of the sub-gaussian regime? Do we have any indication? (Experiments with artificial data could be useful here)
2) Could you evaluate the effect of the sampling on these experiments?

**Limitations:**

Yes

**Strengths And Weaknesses:**

Soundness:

The theoretical development is mathematically rigorous, with clear assumptions (sub-Gaussianity, independence across feature pairs, continuity conditions). The generalized Hanson–Wright inequality is a natural and well-motivated extension that supports the main claims. Results are internally consistent and logically structured: concentration -> geometry -> topology -< isometry. The random function model is sufficiently general to cover non-i.i.d. and heterogeneous settings. However, the reliance on sub-Gaussian assumptions is somewhat restrictive; the paper acknowledges that weaker tails would degrade rates but does not fully characterize this regime. Finite-sample implications of conditions are not sharply quantified, limiting immediate practical interpretation. Some results depend on population-level quantities, and the effect of estimating these in practice is not deeply analyzed.

Presentation:

The paper is well structured, with a clear progression from theory to applications. Key concepts are properly illustrated and the figures are adequate. Some technical sections (e.g., kernel decomposition, normalization issues) may be dense for non-experts.

Significance:

A new view on the curse of dimensionality is highly relevant. The authors provide a reasonable alternative to the classical theory that is compliant with the manifold hypothesis. However, the practical impact depends on whether effective dimension can be reliably estimated and controlled in real applications.

Originality:

To the best of my knowledge, he generalized Hanson–Wright inequality for dependent vectors and the clarification of the explicit role of effective dimension in geometric recovery are new, valuable contributions

---

> ### Author Rebuttal · Authors · 2026-03-30
>
> Thank you for your review and insightful questions, which we address below.
>
> > *``1. What happens out of the sub-gaussian regime? Do we have any indication? (Experiments with artificial data could be useful here)"*
>
> We briefly discuss this in Section 7, noting that weakening the assumptions to sub-exponential or polynomial moment conditions would result in slower convergence rates. We detail this further in Appendix A: if we relax the assumption to a finite fourth moment condition, the convergence rate slows from depending on $\log n$ to a rate of $n /\sqrt{p}$.
>
> To provide a concrete indication of this behaviour we have followed your suggestion and conducted a new experiment using simulated data, which will be included in the revised Appendix (see figure https://ibb.co/Zjyb10z). For a range of distributions with $p=100$, mean-zero and identity-covariance, we plotted the maximum off diagonal inner product ($\max_{i\neq j}|Y_i \cdot Y_j|/p$) as sample size increases (averaged over 10 trials).
>
> > *``2. Could you evaluate the effect of the sampling on these experiments?''*
>
> To quantify the effect of finite sampling on our estimation of $p_\mathrm{eff}$, we will add the simulation in the linked figure (https://ibb.co/GvPV51ZV) to the Appendix. We sample from a fixed Gaussian distribution with $p=100, p_\mathrm{eff}=28$ and show that the estimator $\widehat{p_{\mathrm{eff}}}$ is systematically biased downwards, but asymptotically converges to the true value as $n$ grows. This implies that the estimated values in Table 1 act as conservative lower bounds. We thank the reviewer for highlighting this as it reinforces our claim that the reported values of $\widehat{p_{\mathrm{eff}}}$ are of the same order or higher than $\log n$.

---

> > ### Author Rebuttal · Reviewer_BVaR · 2026-04-01
> >
> > I thank the authors for their reply. I'm happy with the reply to my questions and those of other reviewers, so I will update my score accordingly.

---

### Decision · Program_Chairs · 2026-04-30

**Decision:**

Accept (regular)

**Comment:**

After carefully reading the reviews, the rebuttal and following the subsequent discussion, my assessment is that this paper makes a strong contribution that reviewers generally viewed favorably: the work introduces a statistical framework distinguishing effective dimension, correlation rank, and latent intrinsic dimension; it proves a generalized Hanson–Wright concentration result and uses that to argue that when effective dimension dominates (log?) sample size, pairwise geometry and persistent homology recover latent structure. Across all reviews, the main strengths are originality, the clean conceptual separation of different notions of dimensionality, the bridge from concentration to manifold/TDA recovery, and the interesting neuroscience application. Three reviewers explicitly recommend accept, and the fourth, after rebuttal, raised his/her score to weak accept; furthermore, several reviewers state that their concerns were fully resolved.

For completeness, the criticisms raised by the reviewers are real, but mostly bounded and addressable (and they have been addressed to a large extent in the rebuttal). The main technical reservations concern the dependence on sub-Gaussian and related assumptions, the interpretation of asymptotic or finite-sample concentration rates, and how directly the theory applies to grid-cell data and geodesic-distance computations. There were also some presentation concerns regarding Sections 3,4 being considered too dense; adding some intuition would substantially strengthen the paper and make it more readable.

Due to the ambitious nature of the submission, the strong rebuttal and the fact that all reviewers found the paper novel and important, I am recommending acceptance. I also do encourage the authors to take the remaining concerns seriously (e.g., adding intuition where it fits, sharpening assumptions, explaining applicability, and improving exposition) and incorporate them into their final version.